# Structure of human CALHM1 reveals key locations for channel regulation and blockade by ruthenium red

Johanna L. Syrjänen[1], Max Epstein[1], Ricardo Gómez [1] & Hiro Furukawa [1]✉

Calcium homeostasis modulator 1 (CALHM1) is a voltage-dependent channel involved in neuromodulation and gustatory signaling. Despite recent progress in the structural biology of CALHM1, insights into functional regulation, pore architecture, and channel blockade remain limited. Here we present the cryo-EM structure of human CALHM1, revealing an octameric assembly pattern similar to the non-mammalian CALHM1s and the lipid-binding pocket conserved across species. We demonstrate by MD simulations that this pocket preferentially binds a phospholipid over cholesterol to stabilize its structure and regulate the channel activities. Finally, we show that residues in the amino-terminal helix form the channel pore that ruthenium red binds and blocks.

Large-pore channels conduct ions, osmolytes, and metabolites in various tissues and organs to mediate cellular signaling and homeostasis[1]. CALHM1 is the recently identified member of a large-pore channel that permeates ions, including $Ca^{2+}$, $Na^+$, $K^+$, and $Cl^-$, and small molecules, such as ATP, in a voltage-dependent manner[2]. CALHM1 activity has been demonstrated to modulate neuronal activity and the accumulation level of amyloid-beta[3,4]. Furthermore, the ATP efflux through the CALHM1 channel in type II gustatory cells and the subsequent purinergic signaling has been shown to facilitate the perception of sweet, bitter, and umami taste sensations[5]. Mutations in the *calhm1* gene have been identified as a risk factor for the early onset of Alzheimer's disease in some population groups[3]. A number of structures of CALHM1 and the other CALHMs (CALHM2, 4, 5, and 6) became available by single-particle cryo-EM recently[6–9]. In all cases, the CALHM proteins assemble as multi-oligomers, where CALHM1, 2, 4, 5, and 6 proteins assemble as an 8-*mer*, 11-*mer*, 10-11-*mers*, 11-13-*mers*, and 10-11-*mers*, respectively. These oligomeric states are controlled mainly by interactions between the C-terminal helices (CTHs) of neighboring protomers as well as between transmembrane domains (TMD) 2 and TMD4[1]. Voltage-gated ion channel currents and ATP conductance have been definitively shown for CALHM1 and CALHM1/3 heteromeric channels[2,10]. CALHM2, CALHM4, and CALHM6 have been reported not to form functional channels by several groups[7,9,10]. One study reported a macroscopic current of CALHM2 in the presence of EGTA, which is attenuated by ruthenium red (RuR)[6]. Whether CALHM2 forms functional channels or binds RuR remains inconclusive. Thus far, there are three published cryo-EM structures of CALHM1 channels from non-mammalian species;[9,11–13] however, a number of fundamental questions remain: (1) Do mammalian CALHM1s have similar protomer architectures and oligomeric assembly patterns? (2) What are critical structural motifs that regulate channel activity? and (3) Where is the channel pore and how does ruthenium red (RuR) block the CALHM1 bona fide voltage-dependent channel activity? Here, we address these shortfalls by obtaining the structures of human and chicken CALHM1 channels in the same condition for a direct comparison, and a human CALHM1 potentiating mutant in the presence and absence of RuR. We show the overall structural similarity between the human and chicken CALHM1 channels, a phospholipid-filled hydrophobic pocket critical for structural stability and functions, and the channel pore composed of the aminoterminal helices (NTHs), which RuR can physically block.

## Results

### Structural analysis and comparison of human and chicken CALHM1s

To compare structures and assembly patterns of mammalian and non-mammalian CALHM1 channels, we determined the cryo-EM structure of human CALHM1, as well as that of chicken CALHM1 prepared using an identical protocol (see Methods). Here, we designed the equivalent C-terminal truncation constructs to the killifish CALHM1 (chCALHM1Δct and hCALHM1Δct), previously shown to boost the expression level[11], expressed them in human embryonic kidney (HEK) 293 S GnTI⁻ cells, purified and reconstituted into lipid nanodiscs. Our

[1]W.M. Keck Structural Biology Laboratory, Cold Spring Harbor Laboratory, Cold Spring Harbor, New York, NY 11724, USA. ✉e-mail: furukawa@cshl.edu

single-particle cryoEM resulted in the octameric chCALHM1Δct and hCALHM1Δct structures, resolved at the overall resolutions of 3.36 and 3.76 Å, respectively (Fig. 1a and Supplementary Fig. 1, 2). The structures revealed that the chCALHM1Δct and hCALHM1Δct protomers have a similar protein fold and octameric assembly pattern, where the C-terminal helix (CTH) and TMD4-CTH linker are the major determinants for controlling the oligomeric state (Fig. 1a, b and Supplementary Fig. 3a, b). As is the case for non-mammalian CALHM1s, the most extensive inter-subunit interactions form between the TMD2 and TMD4 and between the CTHs in hCALHM1 (Fig. 1b). In fact, many of the subunit interface residues at TMD2, TMD4, and CTH are conserved among human, mouse, chicken, killifish, and zebrafish orthologues (Fig. 1b, *). It is worth mentioning that we observed octameric and nonameric assembly as assessed by the 'top' views of 2D classification in both chCALHM1Δct and hCALHM1Δct; however, we could only 3D reconstruct the octameric channel as the majority of the particles belong to the octameric species (Supplementary Fig. 3c). Such a mixture of the octameric and nonameric assembly was also observed in the kfCALHM1Δct when we reanalyzed the publicly available dataset (EMPIAR #10444, Supplementary Fig. 3d). In contrast, the full-length chCALHM1 that was trypsinized post-purification displayed only octameric architectures[9]. These observations suggest that the region C-terminal to the CTH (e.g., residues 303-346 in hCALHM1), despite low overall sequence identity among species, may facilitate an octameric assembly during biogenesis.

The overall protomer organization is similar between chCALHM1Δct and hCALHM1Δct (RMSD = 1.016 Å over 186 Cα positions) (Fig. 1c), illustrating the structural conservation despite differences in amino acid sequences (72% identity and 90.6% similarity within CALHM1Δct). Furthermore, chCALHM1Δct and the full-length chCALHM1 (PDB-6VAM; EMD-21143) protomers are structurally similar, indicating the truncation of residues after CTH does not alter protein folds (Fig. 1c). The structures of chCALHM1Δct and hCALHM1Δct are most similar within the TMD regions (RMSD = 0.826 Å; over 140 Cα positions), whereas that of CTH is more variable (RMSD = 1.294 Å; over 49 Cα positions). Our hCALHM1Δct structure allows mapping of key residues for functions and human mutations, Pro86Leu and Arg154His, implicated as risks of Alzheimer's disease[3,14]. The Alzheimer's disease mutation sites, Pro86 and Arg154, are located on the TMD2-3 cytoplasmic loop and the TMD3-4 extracellular loop, respectively (Supplementary Fig. 3e). In our hCALHM1Δct structure, the cryo-EM density was resolved up to Pro85 but not Pro86. However, our chCALHM1Δct map allowed the modeling of residues 85-90 and revealed that these residues are located outside the center pore, indicating that Pro86 is likely not directly involved in the alteration of channel functions. Arg154 is surrounded by residues, Arg158 and Gly139, from the neighboring subunit in the hCALHM1Δct (Supplementary Fig. 3e). It is possible that the presence of a histidine (e.g., pKa of an imidazole ring ~6) in the Arg154His mutant may alter the local structure in a pH-sensitive manner to facilitate signaling for amyloid-beta accumulation as previously reported[3,15]. Additionally, Asp121, which has been reported to modulate calcium sensitivity and ion permeability[2], is located on TMD3 and at the inter-subunit interface with TMD4 from the neighboring subunit to contribute to the structural integrity of the oligomer (Supplementary Fig. 3e).

## Conserved lipid-binding hydrophobic pocket

Inspection of the hCALHM1Δct and chCALHM1Δct structures reveals a hydrophobic pocket ~15 Å in length (parallel to the TMDs) and 10 Å wide, located between TMD3 and TMD4 of one subunit and TMD2 of the neighboring subunit (Fig. 2a). This pocket is formed mostly by the hydrophobic residues, which are conserved among the CALHM1 orthologs (Fig. 2a, Supplementary Fig. 4a). In both hCALHM1Δct and chCALHM1Δct, we observe phospholipid-like density nestled into this pocket, which appears to fortify the local architecture by interacting with the hydrophobic residues in the pocket (Fig. 2a, Supplementary Fig. 4b). A similar hydrophobic pocket in kfCALHM1 was populated with a molecular model of cholesteryl hemisuccinate, a component of which was supplemented during protein purification of this sample[11]. In the current study, no lipid was supplemented during purification; thus, the phospholipid-like molecule is either carried from the expression host cell (HEK293S GnTI-) or the process of the lipid nanodisc reconstitution. The observations above led us to speculate that a phospholipid may be the physiological lipid that occupies and stabilizes the hydrophobic pocket.

To assess our structure-based hypothesis, we estimated the binding stability of a phospholipid, 1-palmitoyl-2-oleoyl-sn-glycero-3-phosphocholine (POPC), and cholesterol by coarse-grained molecular dynamics (CG-MD) simulations on hCALHM1Δct embedded into a membrane bilayer consisting of a 10:3 ratio of POPC to cholesterol (Fig. 2b). During three replicates of the unbiased CG simulations, we observed that each of the eight hydrophobic pockets of the octameric human CALHM1 became populated with a lipid molecule (cholesterol or phospholipid) during the 6 μs timescale of the simulations (Supplementary Fig. 4c, d) corroborating the cryo-EM data that this pocket readily accommodates lipidic molecules. Computation of the Potentials of Mean Force (PMF) in the CG representation reveals that the energy minimum of POPC corresponds to approximately −6.4 kcal/mol, with a clearly defined energy well (Fig. 2b, right panel). In contrast, the cholesterol PMF is comparatively flatter, with an energy minimum of approximately −1.9 kcal/mol (Fig. 2b, right panel), indicating that the binding of a POPC is energetically more favorable than that of cholesterol in this cavity. Thus both the cryo-EM and MD simulations support phospholipid as the preferred physiological lipid in the hydrophobic pocket.

## Alteration of the hydrophobic pocket affects channel functions

To understand the functional role of the lipid-filled hydrophobic pocket, we generated tryptophan point mutations of the pocket forming residues and assessed their functional effects by measuring voltage-gated currents by whole-cell patch-clamp electrophysiology (Fig. 3a, b, Supplementary Fig. 5). Specifically, we measured and compared current density (pA/pF) at +100 mV membrane potential (jumping from −60 mV), where we observed sufficiently large amplitudes for comparative analyses (Supplementary Fig. 5). Amongst the mutants, the functional effects of Val192Trp and Ile109Trp stood out (Fig. 3b). First, the Val192Trp mutation at the deep narrow region of the pocket eliminated current density. Second, incorporation of the tryptophan mutation at the 'entrance' of the hydrophobic pocket (Ile109Trp) exhibited an approximately five-fold increase in current density compared to the wildtype hCALHM1Δct. The other mutants showed no statistically significant changes compared to the wildtype (Fig. 3b).

To delineate whether the upregulation and downregulation by the site-directed mutations are caused by changes in the cell surface expression levels or ion channel activities, we monitored the surface expression levels of the wildtype and mutant hCALHM1Δct. This was first achieved by biotinylation of cell surface hCALHM1Δct proteins at the extracellular lysine (Lys123) by NHS-SS-biotin followed by the pull-down by Strepavidin-Sepharose, elution by DTT, and detection by western blot (Fig. 3c, d). The Ile109Trp mutant showed a similar level of cell surface expression to the wildtype hCALHM1Δct, whereas Ala199Trp showed higher expression level. The Leu67Trp, Val112Trp, Ala116Trp, Thr196Trp mutants showed lower cell surface expression compared to the wildtype hCALHM1Δct and no expression was detected for the Val192Trp mutant in the western blot, indicating instability of the mutant. The Strepavidin-Sepharose eluants of the Ile109Trp, Val192Trp, and Ala199Trp mutants representing robust electrophysiological effects (Ile109Trp and Val192Trp) and no effects but higher expression level (Ala199Trp), along with the wildtype

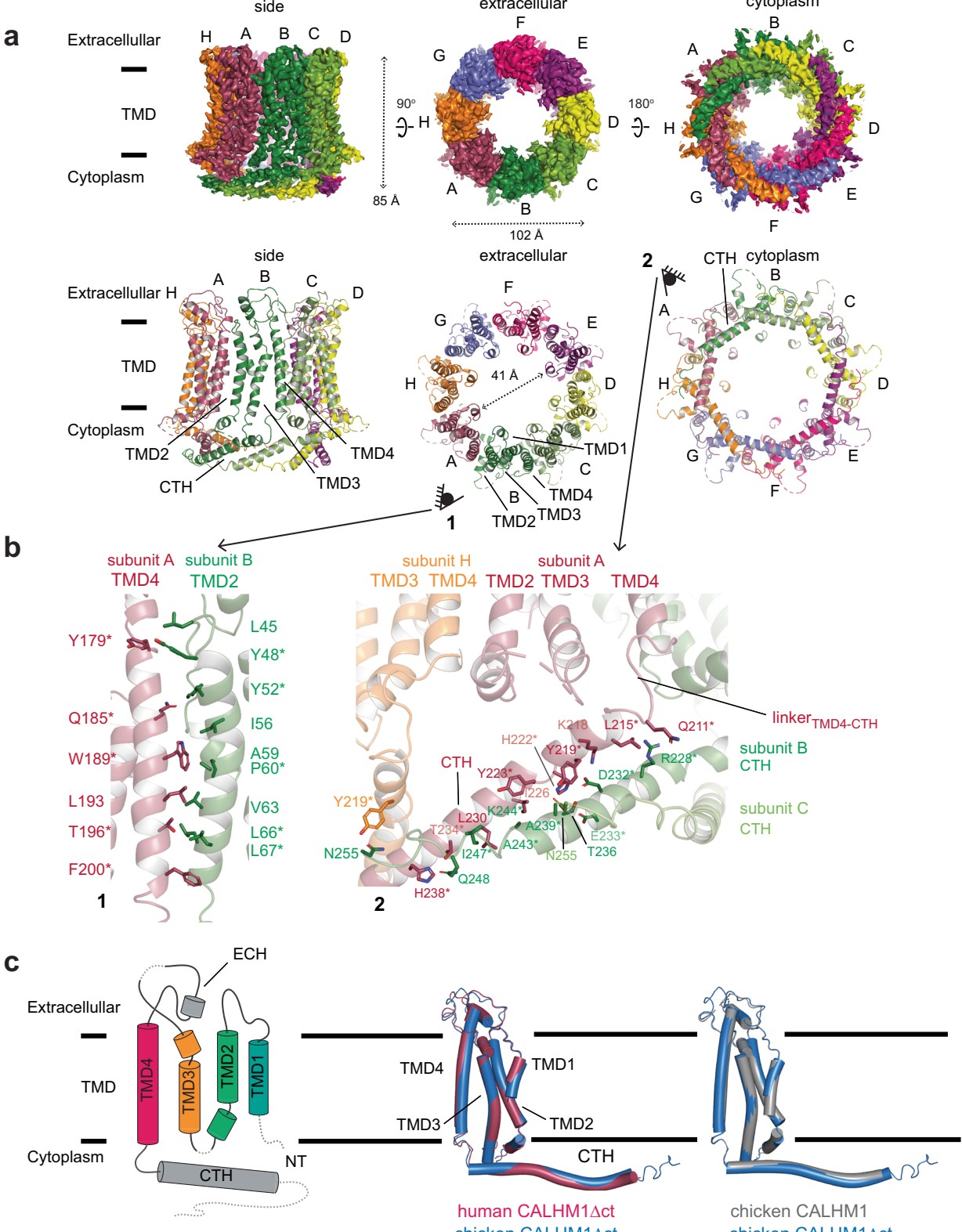

**Fig. 1 | The cryo-EM structure of human CALHM1. a** Cryo-EM density (top) and molecular models (bottom) of human CALHM1 viewed from the side of the membrane, the extracellular region, and the cytoplasm. The pore distance indicated by the double-headed arrow is measured between the Gln33 Cα positions of chains A and E. **b** Inter-subunit interactions between transmembrane helices of human CALHM1 (left). Neighboring subunits are colored red (TMD4 of subunit A) and green (TMD2 of subunit B). Inter-subunit interactions within the C-terminal helix (CTH) are shown (right) across four subunits in orange (subunit H), red (subunit A), green (subunit B), and splitpea (subunit C). Residues conserved across human, mouse, chicken, killifish, and zebrafish are indicated by asterisks. **c** Topology schematic of the human CALHM1 protomer (left) with TMDs, extracellular helix (ECH), and cytoplasmic C-terminal helix (CTH) indicated. The protomer of the chicken CALHM1Δct model superposed with the human CALHM1Δct model (middle) and with the chicken CALHM1 model (right; PBD code: 6VAM).

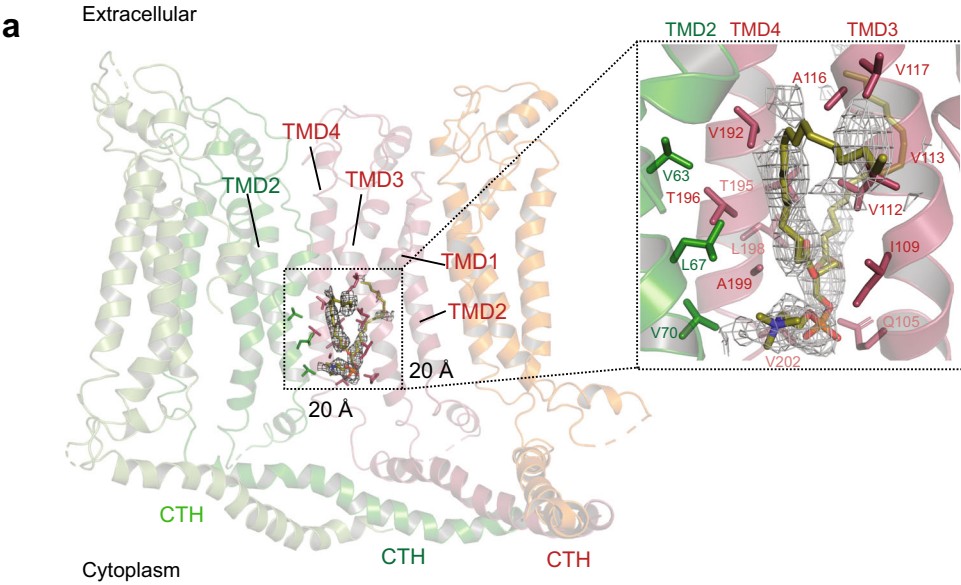

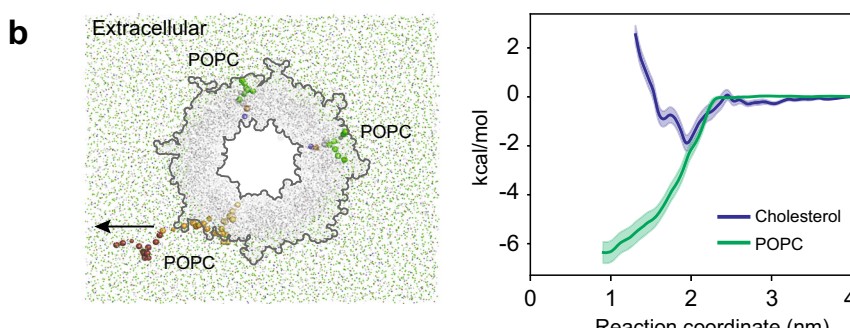

**Fig. 2 | The conserved hydrophobic pocket preferentially binds to phospho-lipid. a** A cross-section of human CALHM1 shows the location of the hydrophobic pocket filled with a phospholipid. Residues that form the hydrophobic pocket are shown in stick representation. There are eight hydrophobic pockets per CALHM1 octamer. The inset shows a zoomed-in view of the hydrophobic pocket viewed from the pore. Residues are numbered in red (TMD3 and TMD4) or green (TMD2). **b** Coarse-grained PMF calculations suggest that binding of POPC into this pocket is thermodynamically favored over that of cholesterol with energy minimums of

approximately −6.4 kcal/mol and −1.9 kcal/mol, respectively (right). The arrow indicates the direction of steering for which the reaction coordinate was generated (left). The steered POPC molecule is colored from yellow to red according to progress along the steered MD simulation, with light yellow beads representing POPC at the start, and red beads at the end of the steered MD. Two other bound POPC molecules are shown in blue, brown and green beads, representing different bead types. Error bands are 1 standard deviation generated from 200 rounds of bootstrap analysis.

hCALHM1Δct, were further analyzed by the antibody-based fluorescence detection size exclusion chromatography (ab-FSEC)[16] (Fig. 3e). This assay measures the fluorescence intensity of the cell surface hCALHM1Δct proteins labelled with anti-1D4 antibody, conjugated to fluorescein isothiocyanate (FITC), at a SEC peak representing folded and oligomerized hCALHM1Δct proteins (Fig. 3c). We observed the fluorescence peak heights with the hierarchy, Ala199Trp > wildtype hCALHM1Δct ≈ Ile109Trp > Val192Trp (no peak) (Fig. 3e), consistent with the western blot analysis (Fig. 3c, d).

Overall, the mutations above alter both channel functions and cell surface expressions, indicating the hydrophobic pocket to be a critical motif for controlling the CALHM1 activities. The impact of mutations on cell surface expression and current density varies depending on their location. Specifically, Ala199Trp causes an increase in surface expression, while Leu67Trp, Val112Trp, Ala116Trp, and Thr196Trp result in decreased expression at different levels, and Val192Trp shows no detectable expression. Regarding current density, Ile109Trp leads to an increase, whereas Leu67Trp, Val112Trp, Ala116Trp, and Thr196Trp exhibit similar levels, and Val192Trp shows no current at all. Comparison of current density with cell surface expression suggests

that the Ile109Trp and Ala199Trp mutations result in upregulation and downregulation of channel activities, respectively (e.g., open probability or conductance level). In contrast, Val192Trp is structurally disruptive, as our cryo-EM structure suggests it could cause steric clashes with Leu120, Trp189, Leu193, and Val63 (Fig. 3a). Furthermore, the Leu67Trp, Val112Trp, Ala116Trp, and Thr196Trp mutants may form upregulated channels.

**Structural analysis of CALHM1-Ile109Trp mutant sheds light on the mechanism of potentiation**

To understand the mechanism of functional upregulation in the CALHM1-Ile109Trp mutant (hCALHM1$_{I109W}$Δct), we conducted single-particle cryo-EM on this mutant. We found that overexpression of the hCALHM1$_{I109W}$Δct in HEK293 cells resulted in low cell survival, likely due to toxicity caused by the upregulated channel activities. This issue was circumvented by adding the channel blocker, RuR, to the culture media, which allowed protein expression and purification. We obtained the structure of hCALHM1$_{I109W}$Δct in lipid nanodiscs and in the presence of RuR at 3.91 Å overall resolution with imposed C8 symmetry (Fig. 4a, b, Supplementary Figs. 6–7). Focused 3D

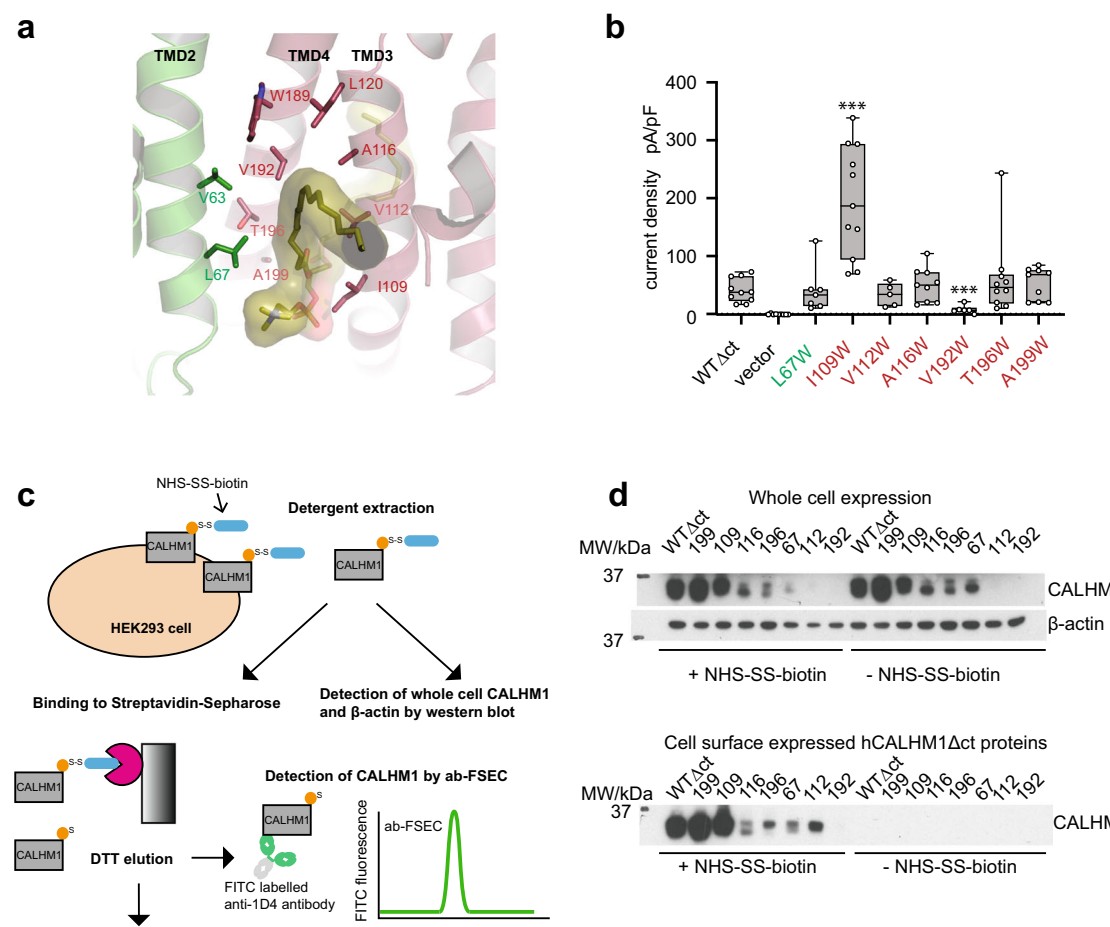

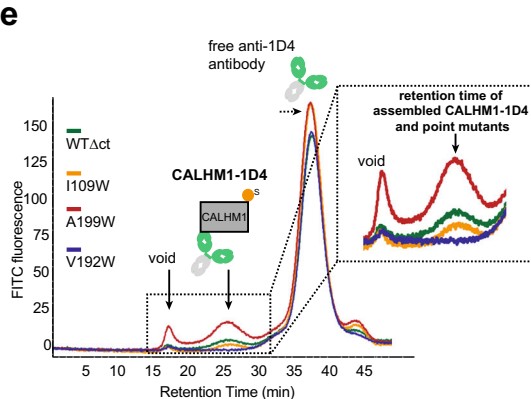

classification around the central pore of this novel structure revealed previously unresolved density of the N-terminal helices (NTHs) and extended density for TMD1, as well as density consistent with RuR, which we will discuss in detail later (Fig. 4a, b, Supplementary Figs. 7–8).

Our structural analysis revealed that in hCALHM1$_{I109W}$Δct, the hydrophobic pocket retains a lipid. Here, the tryptophan residue forms a stronger van der Waals interaction with the lipid than the original isoleucine residue (Fig. 4c). To validate this cryo-EM observation, we performed CG-MD PMF calculations on NTH containing hCALHM1$_{I109W}$Δct and a corresponding homology model where Ile109Trp was reverted to isoleucine, to determine if Ile109Trp strengthens lipid placement compared to hCALHM1Δct. Indeed, these

MD simulations showed that POPC binds more favorably to the hCALHM1$_{I109W}$Δct than hCALHM1Δct (−9.2 kcal/mol in hCALHM1$_{I109W}$Δct vs −5.5 kcal/mol in hCALHM1Δct), supporting the cryo-EM observation that the Ile109Trp mutation stabilizes the POPC binding (Fig. 4d, left panel). The simulations also demonstrated that cholesterol binds more favorably to hCALHM1Δct than hCALHM1$_{I109W}$Δct (+7.2 kcal/mol in hCALHM1$_{I109W}$Δct vs −1.45 kcal/mol in hCALHM1Δct) (Fig. 4d, right panel) but less favorably than POPC, indicating that the Ile109Trp mutation did not change the lipid specificity. Finally, comparison of PMF values between hCALHM1Δct with (Fig. 4d) and without (Fig. 2b) the pore-lining NTHs shows only a small difference in the POPC binding (-0.9 kcal/mol) and cholesterol binding (0.45 kcal/mol) except for the introduction of energy barrier

**Fig. 3 | The conserved hydrophobic pocket is a key locus for structural integrity and channel functions. a** Positions of residues (sticks) within the hydrophobic pocket, analyzed by site-directed mutagenesis. **b** Current density pA/pF at +100 mV for each point mutant (color coded as in *panel a*). Each data point represents a measurement on a different cell (wild-type hCALHM1Δct, $n = 10$; Ile109Trp hCALHM1Δct, $n = 11$; Ala116Trp hCALHM1Δct, $n = 9$; Ala199Trp hCALHM1Δct, $n = 9$; Thr196Trp hCALHM1Δct, $n = 10$; Val192Trp hCALHM1Δct, $n = 6$; vector, $n = 9$; Val112Trp hCALHM1Δct, $n = 5$; Leu67Trp hCALHM1Δct, $n = 7$). Boxes represent the median, 25th, and 75th percentile values, and the whiskers represent the minimum and maximum values. *** denotes $p < 0.001$ versus wild-type. An unpaired two-tailed *t*-test with Welch's correction was used to analyze data. *P*-values are as follows: hCALHM1Δct vs vector, $p = 0.0001$; hCALHM1Δct vs Leu67Trp hCALHM1Δct, $p = 0.9925$; hCALHM1Δct vs Ile109Trp hCALHM1Δct, $p = 0.003$; hCALHM1Δct vs Val112Trp hCALHM1Δct, $p = 0.4962$; hCALHM1Δct vs Ala116Trp hCALHM1Δct, $p = 0.4395$; hCALHM1Δct vs Val192Trp hCALHM1Δct, $p = 0.0006$; hCALHM1Δct vs Thr196Trp hCALHM1Δct, $p = 0.3810$; hCALHM1Δct vs Ala199Trp hCALHM1Δct, $p = 0.3253$. WTΔct is used to refer to hCALHM1Δct in the graph. Source data are provided as a Source Data file. **c** Assessing surface expression of hCALHM1Δct wild-type and selected point mutations. The hCALHM1Δct (C-terminally tagged with the 1D4 epitope) trafficked to the plasma membrane are biotinylated at Lys123 by NHS-SS-biotin, followed by detergent extraction and purification by Streptavidin-Sepharose. The 'pulled-down' human CALHM1Δct is mixed with a FITC-labeled anti-1D4 antibody and analyzed by ab-FSEC for protein quantity (fluorescence intensity) and size (retention time). **d** Representative western blots of hCALHM1Δct wild-type and selected point mutations. The top blot shows samples with and without treatment of NHSSS-biotin, taken after detergent solubilization. The samples were probed with anti-1D4 antibody and anti-β-actin antibody, representing whole cell expression of CALHM1 (human CALHM1Δct) and the β-actin loading control, respectively. The bottom blot shows samples probed with anti-1D4 antibody, representing detection of human CALHM1Δct and point mutant samples expressed at the cell surface with and without treatment of NHS-SS-biotin. This assay was repeated independently with similar results two (Leu67Trp, Thr196Trp) to three (Ile109Trp, Ala199Trp, Val192Trp, Ala116Trp, Val112Trp and hCALHM1Δct) times. Source data are provided as a source data file. WTΔct is used to refer to hCALHM1Δct in the blots. **e** Representative abFSEC traces using a superose 6 10/300 size-exclusion chromatography for the wild-type (WT) and the selected mutants. Peaks representing the void, the hCALHM1Δct- and point mutant-1D4 antibody complexes, and the free 1D4 antibody are observed. The zoom-in view of the hCALHM1-1D4 peak (inset) shows differences in the protein amount.

---

of up to 2.5 kcal/mol for cholesterol binding not present in the NTH lacking model, consistent with no direct involvement of NTHs in lipid binding.

Further inspection of the hCALHM1$_{I109W}$Δct structure provided a potential mechanism underlying the channel functional upregulation. The Ile109Trp is positioned to form stronger interactions with Asn21 and Ala28 on TMD1 than the wildtype, thereby strengthening the interaction between TMD3 and TMD1. The more ordered TMD1 facilitates the interaction between the lipid and Phe19 located in the linker between TMD1 and NTH to contribute to ordering and positioning the pore-forming NTH in the 'upright' manner, as observed in the cryoEM structure (Fig. 4c). Although binding of RuR contributed to further ordering of the NTHs to permit visualization in cryo-EM, we speculate such an NTH conformation occurs in the absence of RuR as well. The 'upright' NTH conformation maximizes the pore size and facilitate ion permeation. In contrast to TMD1 and NTH, little or no change was observed in TMD2-4 and CTH between RuR-hCALHM1$_{I109W}$Δct and hCALHM1Δct (RMSD = 0.787 Å; over 178 Cα positions), indicating that TMD1 and NTH move relative to TMD2-4 and CTH to control ion channel functions.

### Contribution of lipid binding to protein stabilization

Our cryo-EM structures and functional analyses above provided insights into the pattern of lipid binding and its crucial role in channel functions. Indeed, our cryo-EM structures suggested that the binding strength of the integrated phospholipid could control the dynamics of the CALHM1Δct protein. Therefore, to further investigate the contribution of lipid binding to overall protein stability, we conducted all-atom unbiased MD simulations with either POPC ($8 \times 100$ ns), cholesterol ($9 \times 100$ ns) or in an apo, non-lipid bound state ($9 \times 100$ ns) in each of the eight hydrophobic pockets on the NTH containing hCALHM1Δct homology model. The population density of Cα atom RMSDs (Fig. 5) for the apo (red), cholesterol-bound (green) and POPC-bound (blue) structures demonstrate a rank order of stability of POPC > cholesterol > apo, with mean values of 5.64 Å, 5.79 Å and 5.95 Å respectively. Comparison between the POPC and apo distributions show a shift in the population densities with larger deviations for the apo state. It is likely that further divergence in the magnitude of these changes occur with longer timescale simulations. These data are thus commensurate with the identification of this cavity as an obligate lipid binding pocket. Furthermore, the cryo-EM structures and the MD simulations collectively demonstrate the vital role played by the lipid within the hydrophobic pocket in maintaining the structural integrity of the CALHM1 channel.

### Structural analysis of CALHM1-Ile109Trp reveals the RuR binding site in the channel pore

The single-particle cryo-EM on hCALHM1$_{I109W}$Δct in the presence of RuR showed oblong density consistent with the size of RuR in the central pore surrounded by the NTHs (Fig. 6a and Supplementary Fig. 8). The resolution of the cryo-EM map is lower around the NTHs and pore region compared to the TMDs (Supplementary Fig. 8c, f), possibly suggesting a number of subtly different binding modes within this binding pocket since the pore diameter is large. Nevertheless, the cryo-EM density is of sufficient quality for the placement of RuR molecule, although an atomic-resolution pose cannot be determined from these data. This density was not observed in the cryo-EM structure of hCALHM1$_{I109W}$Δct in the absence of RuR, supporting the view that it represents RuR (Supplementary Figs. 9 and 10). The structures of hCALHM1$_{I109W}$Δct in the presence and absence of RuR are similar in TMD2-4 and CTH (RMSD = 0.541 Å; over 197 Cα positions), indicating that the binding of RuR only affects the structure of the central pore comprised of TMD1 and NTH (Supplementary Fig. 11). Moreover, the structure of hCALHM1$_{I109W}$Δct in RuR-free conditions does not clearly resolve the NTHs, indicating a higher degree of conformational flexibility of the NTHs in the absence of RuR. We therefore suggest that the TMD1-Ile109Trp interactions, lipid-Phe19 interactions, and RuR-NTH interactions all contribute towards NTH conformational stabilization.

The RuR molecule is positioned to plug the pore through associations with the NTHs (Fig. 6a). To validate this structural observation, we mutated the pore-lining residues proximal to RuR to arginine (Gln10Arg, Gln13Arg, and Gln16Arg) and assessed RuR-mediated channel blockade by patch-clamp electrophysiology (Fig. 6b–d). We reasoned that arginine residues that face the pore of the channel, being positively charged and bulkier than glutamine, would disfavor binding of the positively charged RuR and decrease the extent of channel blockade. Indeed, the Gln10Arg, Gln13Arg, and Gln16Arg mutants all decreased the extent of channel blockade to different extents (Fig. 6d). Thus, the experiments above validated our structural observation that RuR binds and plugs the central pore formed by the NTHs.

## Discussion

Our studies here showed that the human and chicken CALHM1s have a conserved structural fold, oligomeric assembly pattern, and a phospholipid-filled hydrophobic pocket for controlling channel functions. Site-directed mutation of the hydrophobic pocket residues resulted in alteration of the channel activity. Furthermore, we also revealed that the channel blocker RuR binds the channel pore formed by the NTHs.

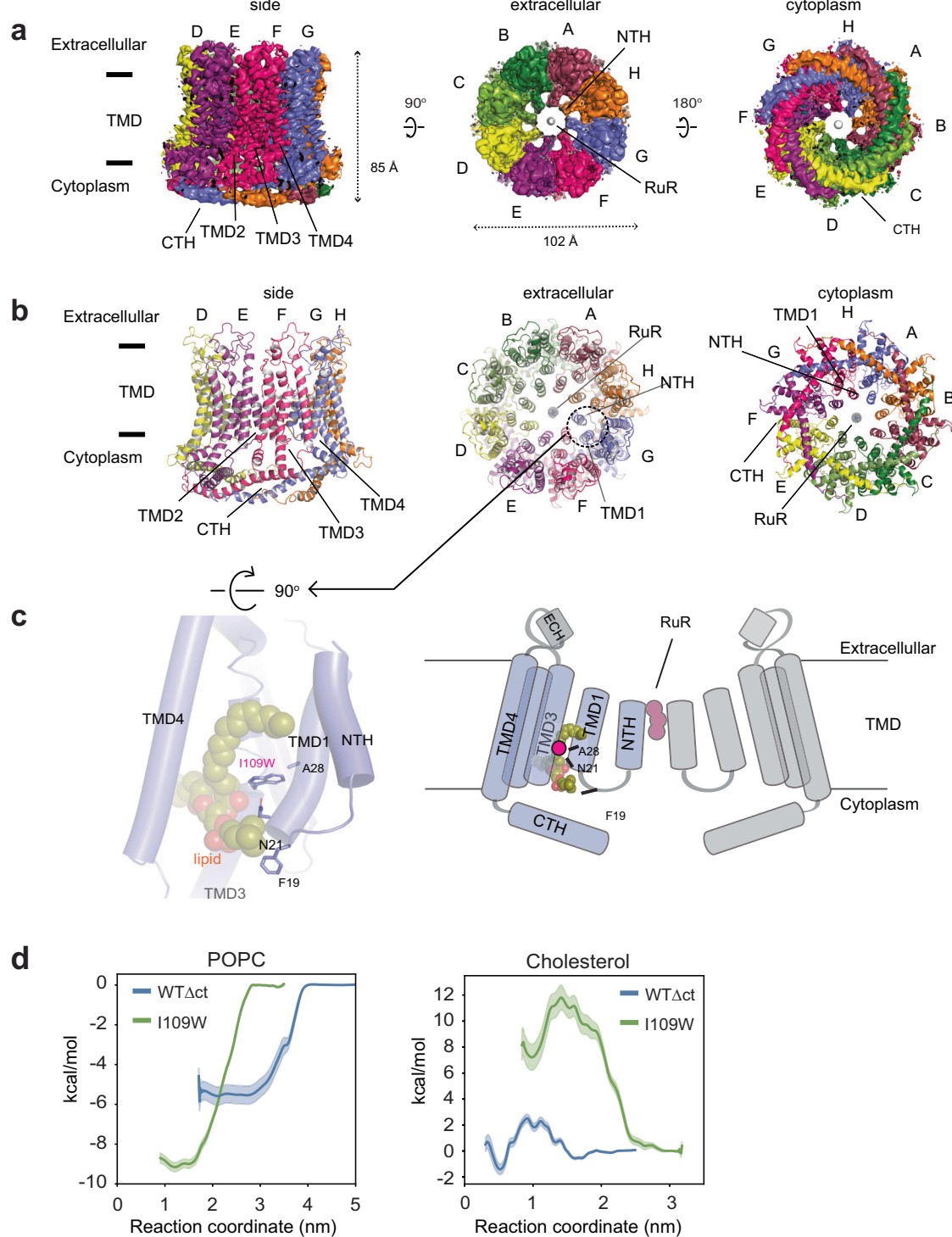

**Fig. 4 | The cryo-EM structure of Ile109Trp hCALHM1WTΔct (hCALHM1$_{I109W}$Δct) in the presence of RuR. a, b** The C8 symmetry imposed cryo-EM density (**a**) and the molecular model (**b**) of RuR-hCALHM1$_{I109W}$Δct viewed from the side of the membrane, the extracellular region, and the cytoplasm. **c** A zoomed-in view of subunit G showing the phospholipid (cryo-EM density and sticks), Ile109Trp, and the residues interacting with them (sticks; left). A schematic representation highlighting the positions of TMD1, the NTH, Ile109Trp (W), Phe19 (F), Asn21 (N), Ala28 (A; black lines) (right). The Ile109Trp mutation stabilizes the interaction between TMD3 and the pore composed of TMD1 and NTH via direct and phospholipid-mediated interactions. **d** PMF calculations of cholesterol or POPC in hCALHM1Δct (WTΔct, blue) or hCALHM1$_{I109W}$Δct (I109W, green). Both models contained NTHs without RuR. The hCALHM1Δct model was built based on the RuR-hCALHM1$_{I109W}$Δct model but with I109W reverted to isoleucine. Bulk phase is reached at different points along the reaction coordinates for respective lipids, as initial steered MD was propagated along the *x*-axis along the bilayer normal from binding pockets located at different rotational angles in the simulation box. Error bands are 1 standard deviation generated from 200 rounds of bootstrap analysis.

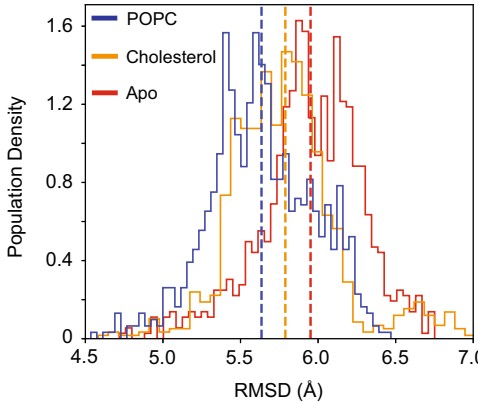

**Fig. 5 | All atom MD simulations of hCALHM1WTΔct in the presence and absence of lipids.** Population density of Cα atom RMSDs for the hCALHM1Δct homology model based on the RuR-hCALHM1$_{I109W}$Δct model with either cholesterol (orange) or POPC (blue) bound to each hydrophobic pocket or in apo state (red). Means of corresponding histograms are plotted as vertical dashed lines of corresponding color. Histograms are each comprised of 50 bins each.

Lipid binding contributes to structural integrity and functional regulations in many membrane proteins. Examples include the Pannexin1 and LRRC8A channels, where lipids enhance the inter-subunit packing[17–19], and Piezo1, where a lipid binding pocket may allosterically regulate the channel activity[20]. Our cryo-EM study showed the presence of the specific binding pocket at the cytoplasmic half of the TMD3 and TMD4, where a phospholipid can bind. Our MD simulations showed that the hydrophobic pocket binds a phospholipid more favorably than cholesterol and that the phospholipid binding stabilizes the CALHM1 channel structure (Fig. 4d; Fig. 5; Supplementary Fig. 12). The Ile109Trp mutation that upregulates the channel activity (Fig. 3) binds phospholipids more favorably at the hydrophobic pocket, indicating that the coupling between the phospholipid binding and channel activity. The lipid-like density between TMD3 and TMD4 was also observed in the cryo-EM maps of CALHM2, CALHM4, and CALHM5 at a similar site, suggesting the lipid-mediated structural stabilization as a common feature amongst most CALHM family members[7–9].

The current study resolved the NTHs complexed with RuR. Our hCALHM1 structure shows that NTHs are positioned approximately orthogonal to the membrane plane and that RuR binds to the pore generated by the NTHs from the eight subunits (Figs. 4 and 6). This NTH orientation maximally sets the pore-diameter to be >14 Å between fully extended side chains (e.g., Gln13 - the side chain density is not resolved), which is sufficiently large to conduct ATP and ions[21]. We speculate that closure of the channel is mediated by the highly dynamic nature of the NTHs, which disrupts the NTH pore architecture in a stochastic manner, as the recent MD simulations study indicated[22]. The architecture and motion of the TMD1-NTH region are analogous to that of the arm, elbow joint, and forearm, with the NTH ("forearm") permitted a range of motion. The Ile109Trp mutation and RuR together appear to stabilize the positioning of TMD1 and NTH, allowing us to resolve the RuR-blocked pore (Fig. 4c, right panel). Despite no structural similarity with CALHM1, the critical role of the NTHs extends to other large-pore channel family members such as pannexin1[23] and connexin[24]. It is worth mentioning that the cryo-EM density of CALHM1 NTH extended toward Asp121, a residue shown to be critical for gating and calcium sensitivity[2]. It is plausible that calcium binding at Asp121 may affect the channel activity by controlling the orientation of the proximally located NTHs.

Finally, our structure on the CALHM1-RuR complex demonstrates the inhibitory mechanism of RuR to be the physical plugging of the NTH pore. It is known that RuR also inhibits other ion channels, including transient receptor potential (TRP), ryanodine, Piezo, and two-pore domain potassium (K2P) channels, and the calcium mitochondrial uniporter[2,6,25–29], but in different binding sites. For instance, RuR binds and inhibits K2P at the extracellular domain at the channel entrance prior to the selectivity filter gate[30]. In contrast, RuR inhibits TRPV6 channels by interacting with the backbone carbonyl oxygens of the ion selectivity filter at TMD like a cork in a bottle[31]. CALHM2 is reported to bind RuR at the top of each TMD1 helix next to residue Phe39. However, the validity of the proposed binding site remains inconclusive since the similar cryo-EM density assigned as RuR is present in the CALHM2 gap junction structure without RuR as well[1,6]. Furthermore, whether CALHM2 forms a channel or not is in debate at this point. Nevertheless, the RuR binding to hCALHM1 is similar to that in TRPV6 in that RuR binds and plugs the TMD pore. The RuR binding in CALHM1 is not as defined as in TRPV6 because of the large diameter of the CALHM1 pore. It may be possible that the presence of RuR in the CALHM1 pore creates a highly charged electrostatic barrier that prevents the flux of ions.

## Methods

### Expression and purification of CALHM1Δct proteins

Chicken CALHM1Δct (residues 1-291 fused to the C-terminal Strep-II tag), human CALHM1Δct (residues 1-303 fused to the C-terminal Strep-II tag), and human CALHM1Δct-Ile109Trp (also C-terminally Strep-II tagged), were cloned into the pFWgfpCMV, the baculovirus transfer vector harboring sequences for the CMV promoter, the WPRE 5' UTR sequence, and EGFP, modified from pFp10[16]. Following bacmid generation, Sf9 cells were used to amplify the recombinant baculovirus. HEK293 GnTI- cells were cultured in FreeStyle 293 expression media supplemented with 2% FBS and at 37 °C. Cells were infected with baculovirus at a density of $4 \times 10^6$ cells/ml at a v:v ratio of 1:20. Following infection, the cultures were moved to 30 °C and supplemented with 5 mM sodium butyrate and 20 µM RuR (for the hCALHM1$_{I109W}$Δct and RuR-hCALHM1$_{I109W}$Δct samples), and harvested 48 h after infection. The cell pellets corresponding to expressed chicken CALHM1Δct were resuspended in 20 mM Tris pH 8.0, 150 mM NaCl, 1 mM PMSF and then C12E8 (Anatrace) was added to a final concentration of 1%. Following 2 h of solubilization at 4 °C, the lysate was clarified by two rounds of centrifugation: the first one at 4,550 g for 20 min (4 °C), and the second at 186,000 g for 1 h at 4 °C. The clarified supernatant was loaded onto a Strep–Tactin Sepharose column (IBA) followed by 20 column volumes of washing with 20 mM Tris-HCl pH 8.0, 150 mM NaCl, 0.01% C12E8 (wash buffer) and elution using the wash buffer supplemented with 3 mM desthiobiotin. The purified chicken CALHM1Δct was concentrated to ~3.9 mg ml−1 at 4 °C using 100-kDa MWCO Amicon concentrators (Millipore) before reconstitution into nanodiscs. For reconstitution into nanodiscs, soybean polar extract, MSP2N2 and the purified chicken CALHM1Δct protein, at final concentrations of 0.75, 0.3, and 0.4 mg/ml, respectively, were mixed for 1 h at 4 °C, followed by detergent removal by SM2 Bio-Beads (BioRad) overnight (~12 h). MSP2N2 was expressed and purified as previously described[32]. The nanodisc reconstitution reactions were pooled and concentrated and run over SEC using a Superose 6 10/300 (Cytiva) in 20 mM Tris pH 8.0, 150 mM NaCl. Peak fractions were pooled and concentrated to ~0.5–2.3 mg/ml, supplemented with 1 mM EDTA. To remove RuR from the hCALHM1$_{I109W}$Δct sample, the Strep-Tactin resin bound protein was washed first in buffer supplemented with 5 mM EGTA pH 7.0, then 20 mM CaCl₂, and finally 40 mM EDTA pH 8.0, followed by extensive washing in 20 mM HEPES-NaOH pH 7.5, 200 mM NaCl. The RuR-hCALHM1$_{I109W}$Δct sample was supplemented with 100 µM RuR after loading on the Strep-Tactin column. The purified hCALHM1Δct, RuR-free hCALHM1$_{I109W}$Δct, and RuR- hCALHM1$_{I109W}$Δct complex samples were concentrated to ~1.0-3.2 mg ml⁻¹ at 4 °C using 100-kDa MWCO Amicon concentrators (Millipore) before reconstitution into nanodiscs. For reconstitution into nanodiscs, soybean polar

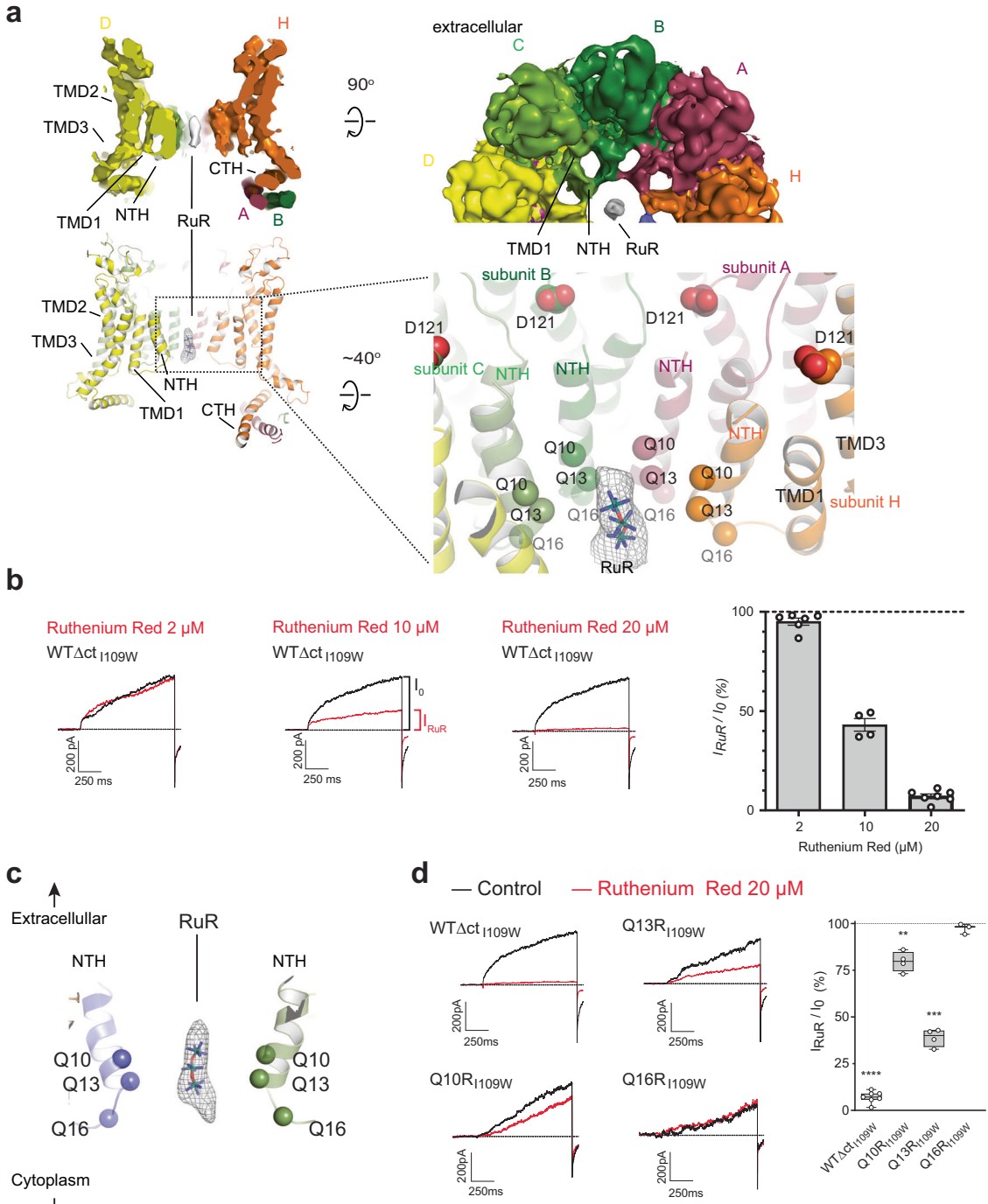

**Fig. 6 | Ruthenium red binding site in the central pore. a** A cross-section of the cryo-EM map (top) and model (bottom) of hCALHM1$_{I109W}$Δct without imposing symmetry (c1). The RuR density is shown as white surface (top) and gray mesh (bottom). The Cα atoms of the pore-lining residues, Gln10, Gln13, and Gln16, are shown as spheres. Asp121 is in the vicinity of NTH. **b** Representative current traces and the concentration-response of the hCALHM1$_{I109W}$Δct channel blockade by RuR at +60 mV (I$_{RuR}$/I$_0$) displayed as a bar chart. Data are represented as individual points, the bars show the mean and the whiskers indicate the standard error of the mean ($n$ = 6, 4, and 7 cells for 2, 10, and 20 μM RuR application, respectively). Definitions of currents without RuR (I$_0$) and with RuR (I$_{RuR}$) are indicated in the middle trace. WTΔct$_{I109W}$ is used to refer to hCALHM1$_{I109W}$Δct in the chart and traces. Source data are provided as a source data file. **c** A cartoon of the central pore, highlighting the positions of Glu10, Glu13, and Glu16 (spheres) around the RuR binding site (cryo-EM density for RuR shown as mesh; RuR model in stick representation). **d** Representative current traces in the absence (black traces) and presence (red traces) of 20 μM RuR at +60 mV for the Glu10Arg, Glu13Arg, and Glu16Arg point mutants. The graph shows the extent of the channel blockade (I$_{RuR}$/I$_0$) calculated from the recordings. Boxes represent the median, 25th, and 75th percentile values, and the whiskers represent the minimum and maximum values ($n$ = 7, 4, 4, and 3 cells for hCALHM1$_{I109W}$Δct, Glu10Arg, Glu13Arg, and Glu16Arg, respectively). ****, ***, and ** denote $p < 0.0001$, $p < 0.001$, and $p < 0.01$, respectively, versus basal conditions (absence of RuR) for each construct studied (two-tailed paired $t$-test: $p < 0.0001$ for WT, $p = 0.0046$ for Glu10Arg, $p = 0.0001$ for Glu13Arg, and $p = 0.2369$ for Glu16Arg). WTΔct$_{I109W}$ is used to refer to hCALHM1$_{I109W}$Δct in the chart and traces. Source data are provided as a source data file.

extract, MSP2N2 and the purified human CALHM1 protein, at final concentrations of 0.75, 0.3 and 0.4 mg ml−1, respectively, were mixed for 1 h at 4 °C, followed by detergent removal by SM2 BioBeads (BioRad) overnight (-12 h). The beads were removed and the solution was further purified by size-exclusion chromatography using a Superose 6 10/300 column (GE Healthcare) in 20 mM HEPES-NaOH pH 7.5, 200 mM NaCl. The buffer for hCALHM1Δct and hCALHM1$_{I109W}$Δct was supplemented with 1 mM EDTA and the RuR- hCALHM1$_{I109W}$Δct sample was supplemented with 50 µM RuR and 1 mM EGTA. Peak fractions were pooled and concentrated to -0.5-1.3 mg/ml for cryo-EM grid preparation.

## Cryo-EM image acquisition and data processing

The chicken CALHM1Δct in nanodisc (3.5 µl), RuR-free hCALHM1$_{I109W}$Δct in nanodisc (4 µl) and RuR-hCALHM1$_{I109W}$Δct in nanodisc (4 µl) complexes were applied to glow-discharged gold grids (R1.2/1.3 UltrAuFoil 300 mesh). The grids used for the hCALHM1$_{I109W}$Δct in nanodisc samples had been treated with 0.01% β-OG immediately beforehand. The human CALHM1Δct in nanodisc (3.5 µl) was applied to 1.2/1.3 400 mesh C-flat carbon-coated copper grids (EMS). Grids with human and chicken CALHM1Δct were blotted for 4 s with blot force 7 at 85% humidity and 15 °C before plunge freezing into liquid ethane using a Vitrobot Mark IV (ThermoFisher). Grids with the RuR-free and RuR-hCALHM1$_{I109W}$Δct samples were blotted for 1.8 s at 85% humidity and 20 °C before plunge freezing into liquid ethane using an EM GP (Leica Microsystems). Datasets were collected using a Titan Krios operated at an acceleration voltage of 300 keV and the GATAN K3 direct electron detector coupled with the GIF quantum energy filter (Gatan) controlled by EPU v2.10.0.5 software (ThermoFisher). Movies were recorded with a pixel size of 0.856 Å, an exposure time of 1.8 s over 30 frames and a dose rate of 2 e − /Å²/frame. For the human and chicken CALHM1Δct data sets, the program Warp (v1.0.9) was used to align movies, estimate the CTF and pick particles[33]. Two-dimensional classification was performed using cryoSPARC (v3) software and ab initio 3D map generation, 3D refinement, 3D classification were performed using Relion 3.1 software[34,35]. Per-particle CTF refinement and B-factor sharpening were then performed using cis-TEM (v1.0.2) software[36]. For the RuR-free and RuR-hCALHM1$_{I109W}$Δct datasets, cryoSPARC was used to align movies, estimate the CTF, and pick particles as well as perform two-dimensional classification and ab initio 3D map generation, 3D refinement, and 3D classification[34]. Focused 3D classification was performed using Relion 3.1 software[35], prior to nonuniform refinement using Cryosparc. Modeling was performed in the programs UCSF Chimera (v1.14) and Coot (v0.893), using the published structure of chicken CALHM1 as a starting model (PDB code: 6VAM)[37]. The final models were refined against the cryo-EM maps using PHENIX (v1.14) real-space refinement with secondary structure and Ramachandran restraints[38]. The FSCs were calculated by phenix.mtriage or within cryoSPARC[34,38]. Local resolutions were estimated using the program ResMap (v1.1.4)[39]. Superpositions of atomic models and subsequent RMSD calculations were performed in Pymol v2.5 using the align algorithm over Cαs unless otherwise specified[40]. Data collection and refinement statistics are summarized in Supplementary Table 1.

## Electrophysiology of human CALHM1Δct and CALHM1I$_{109W}$Δct point mutants

The point mutants Ala116Trp, Ala199Trp, Leu67Trp, Ile109Trp, Thr196Trp, Val112Trp, and Val192Trp on hCALHM1Δct, and Gln10Arg, Gln13Arg, and Gln16Arg on CALHM1$_{I109W}$Δct, Cterminally fused to the 1D4-tag were subcloned into the pFWgfpCMV vector. Adherent HEK293 cells in DMEM + 10 % FBS were transfected (using TransIT-2020, Mirus Bio) with the plasmids, and the cells were analyzed after ~36–48 h. Electrophysiological recordings were obtained from the transfected HEK293 cells using the whole-cell configuration of the

patch-clamp technique[41] at room temperature (22–24 °C). Patch pipettes were made from 1.50 OD/0.86 ID borosilicate glass capillaries (BF150-86-10; Sutter Instruments) using a P-97 micropipette puller (Sutter Instruments) and polished with an MF-830 microforge (Narishige) to a final resistance of 2–6 MΩ when backfilled with the internal solution. The pipette solution for the Ala116Trp, Ala199Trp, Leu67Trp, Ile109Trp, Thr196Trp, Val112Trp, and Val192Trp hCALHM1Δct was (in mM) 147 NaCl, 10 EGTA and 10 HEPES, pH 7.0 with NaOH, and for the Gln10Arg, Gln13Arg, and Gln16Arg CALHM1$_{I109W}$Δct mutants contained (in mM) 110 Cs-gluconate, 30 CsCl, 5 HEPES, 5 BAPTA, 4 NaCl, 2 MgCl$_2$, 0.5 CaCl$_2$, 2 ATP-Na, 0.3 GTP-Na; pH 7.35 with CsOH. Cells were held at −60 mV and superfused with a bath solution, containing (in mM): 150 NaCl, 3 KCl, 10 HEPES, 1 CaCl$_2$, and 0.01 EDTA-Na, at pH 7.4 with NaOH (for Gln10Arg, Gln13Arg, and Gln16Arg CALHM1$_{I109W}$Δct and CALHM1$_{I109W}$Δct in RuR inhibition experiments) or 147 NaCl, 13 glucose, 2 KCl, 2 CaCl$_2$, 1 MgCl$_2$, 10 HEPES, pH 7.3 with NaOH (for Ala116Trp, Ala199Trp, Leu67Trp, Ile109Trp, Thr196Trp, Val112Trp, and Val192Trp hCALHM1Δct hydrophobic pocket mutants). Data were acquired and lowpass-filtered with an AxoPatch 200B patch-clamp amplifier (Molecular Devices). Signals were fed to a Pentium-based PC through an Axon Digidata 1550B interface board (Molecular Devices). hCALHM1$_{I109W}$Δct currents for all the constructs were elicited by application of 1 s steps from −100 mV to +100 mV in 20 mV increments, measured at the end of the step pulse, and normalized to cell capacitance. RuR (2, 10, and 20 µM) was applied using a fast perfusion system (RSC-200; Bio-Logics) until reaching the steady-state of the effect and hCALHM1$_{I109W}$Δct amplitudes at the end of the +60 mV depolarizing step were measured before (control) and after drug application. The pClamp software (Clampex v11.2, ClampFit v11.2; Molecular Devices) was used for stimulus generation, data display, acquisition, storage, and analysis. GraphPad Prism 9 software was used for visualization and statistical analysis. Paired or unpaired (with Welch's correction) two-tailed ttests were used to assess the statistical significance of RuR (control vs. drug), RuR concentration-response, or comparison between WT and single-point mutants, (Supplementary Tables 2–4).

## Surface expression assay of human CALHM1Δct point mutants

HEK293 cells transiently expressing human CALHM1Δct single point mutants were resuspended in cold PBS. EZ-Link Sulfo-NHS-SS-Biotin (ThermoFisher) or a corresponding volume of buffer was added to half of each sample. The samples were nutated at 4 °C for 1 h. Tris-HCl pH 8.0 was added to a final concentration of 50 mM to quench the reaction. Following a wash step in PBS, the cells were lysed in 1% C12E8 (30 min nutation at 4 °C) and the lysate clarified by centrifugation (21,130 g, 15 min, 4 °C). Samples of total cell input were analyzed by western blot. Streptavidin slurry (Sigma) was added and the lysate was incubated under rotation for 2 h at 4 °C. The resin was then washed and incubated in buffer supplemented with 100 mM DTT at room temperature for 2 h to elute bound protein. The elution was analyzed by FSEC and by western blot. For FSEC analysis, 50 µl of eluate was mixed with 1 µl of 0.06 mg/ml of anti-1D4FITC conjugated antibody prior to injection on a Superose 6 10/300 column equilibrated in 20 mM Tris-HCl pH 8.0, 200 mM NaCl, 1 mM EDTA, 0.01% C12E8 running at 0.4 ml/min with excitation/emission wavelengths at 494/521 nm. Samples were subjected to western blot using monoclonal anti-1D4 (University of British Columbia) and anti-mouse horseradish peroxidase-conjugated antibody (Amersham) using dilutions of 1:5000 (of a 1 mg/ml anti-1D4 stock) and 1:20000, respectively in 0.5% nonfat dry milk powder, 20 mM Tris pH 8.0, 150 mM NaCl, 0.05% Tween. In addition, input samples of whole cell content were also probed using mouse monoclonal horseradish peroxidase-conjugated anti-β-actin antibody (Proteintech) at a dilution of 1:20000 in 5% nonfat dry milk powder, 20 mM Tris pH 8.0, 150 mM NaCl, 0.05% Tween. Protein bands were detected by enhanced chemiluminescence on X-ray film (ECL kit; GE Healthcare). Normalization using β-actin was used to compare

protein expression levels between mutants. Two (Leu67Trp, Thr196Trp) or three (Ile109Trp, Ala199Trp, Val192Trp, Ala116Trp, Val112Trp and hCALHM1Δct) independent biological replicates were analyzed by western blot.

## Coarse-grained molecular dynamics simulations

Missing heavy atoms and hydrogen atoms of the cryo-EM determined structure were added using modeller v.10.1[42]. Modeller v.10.1 was also used to generate the hCALHM1Δct system that included the NTHs by reverting Trp109 from RuR-hCALHM1$_{I109W}$Δct to isoleucine. RuR was not included in the simulations. Gromacs 2021.2 pdb2gmx[43] was used to protonate the structures at a pH of 7.4. The resulting atomistic models were converted into MARTINI coarse-grained models using the martinize v2.6 script[44] and embedded into a membrane bilayer with a 10:3 ratio of POPC to cholesterol, solvated with water and a NaCl concentration of 0.15 M with additional net-charge neutralizing ions using the insane.py script[45]. The full coarse-grained system was subsequently energy minimized with the steepest descent algorithm. Tolerance was set to a value of 10 kJ/mol/nm, with an initial step size of 0.01 nm. Restraints were applied to all protein beads with a force constant of 1000 kj/mol/nm². Following energy minimization, the system was equilibrated in the isothermal-isobaric ensemble for 2 ns with a timestep of 10 fs and harmonic restraints applied to all protein beads with a force constant of 1000 kj/mol/nm². This was followed by a second round of equilibration for 2 ns where restraints were maintained on protein backbone beads only. The neighbor list was generated with a Verlet cut-off scheme and updated every 20 timesteps. Verlet-buffer tolerance was 0.005 Kj/mol/ps. Coulombic interactions were treated with reaction-field and a 1.1 nm cut-off. The relative dielectric constant was set to 15 and 0 for the reaction field. The Van der Waals cut-off was set to 1.1 nm with a potential-shift-verlet modifier. The system was coupled to a heat-bath by using the v-rescale temperature coupling algorithm. Temperature was set to 323 K. Pressure was maintained with the Berendsen barostat at 1 bar with semi-isotropic coupling and a 12 ps coupling constant. During production runs, restraints were applied to backbone protein beads with a force constant of 1000 kj/mol/nm². Production runs were 4.2 μs, 5.5 μs and 4.3 μs respectively with a timestep of 0.02 ps for the hCALHM1Δct system that included NTHs (built as a homology model based on the RuR-hCALHM1$_{I109W}$Δct structure). Three 6 μs simulations were performed for the NTH lacking hCALHM1Δct system which were used to spawn initial umbrella sampling configurations. Temperature coupling was the same as specified for equilibration steps but with Parrinello-Rahman pressure coupling. A pressure of 1 bar and coupling constant of 12 ps was maintained. All simulations performed herein were done using Gromacs version 2021.2.

## PMF calculations

Potential of mean force calculations for cholesterol were instigated from a coarse-grained conversion of the cholesterol bound atomistic model via martinize. Initial umbrella windows were generated by steering a single cholesterol molecule laterally across the bilayer along the x-axis with a force constant of 1000 kJ/mol/nm. Windows were initially spaced ~0.5 Å apart using a python script written by Dr. Owen Vickery (DOI: 10.5281/zenodo.3592318) with additional windows placed at undersampled regions. Each window was simulated for a total of 1 μs, with the first 200 ns discarded as equilibration. Each window was restrained with an umbrella potential of 1000 kJ/mol/nm apart from certain windows at saddle points on the free energy surface, where the force constant was increased to 2000 kJ/mol/nm. A temperature of 323 K was used for umbrella sampling windows. All other physical parameters were the same as for unbiased CG simulations. POPC PMFs were initiated from a representative frame of an unbiased CG simulation for the respective system wherein POPC had bound to the putative binding site. One POPC molecule was selected and steered

in the x-dimension, laterally along the plane of the bilayer as depicted in Fig. 2b. The chosen reaction coordinate represented the distance between the backbone bead of residue Gln33 (TMD1) and terminal alkyl bead C4A for POPC and the center of mass of cholesterol. The free energy surfaces were calculated using the weighted histogram analysis method with 200 bootstraps. Convergence was assessed by block analysis (Supplementary Figs. 4 and 12), with each PMF consisting of an additional 200 ns of data. The differences between 600 and 800 ns PMFs were within thermal energy for both POPC and cholesterol.

## All-atom molecular dynamics simulations

The fixed atomistic structure described earlier as the input for coarse grained simulations, was embedded in a pure POPC bilayer using the inflategro.pl script[46]. Eight cholesterol molecules were modeled into the EM density using Coot v 0.893 and Phenix v1.14[37,38]. For the POPC bound system, a single POPC molecule was docked into the proposed binding pocket using Vina and then duplicated for each of the eight potential binding sites using Phenix 1.2[47]. The top scoring pose in which the phosphocholine moiety was positioned towards the pore was selected as this represented the main binding mode seen consistently during unbiased CG simulation (Supplementary Fig. 4 and 12). The system was solvated with TIP3P water and neutralizing Na$^+$ and Cl$^-$ ions added followed by an addition of 150 mM NaCl[48]. The Amber99SB-ILDN forcefield was used for the protein, with amber S-lipid parameters for POPC and cholesterol[49–52]. Periodic boundary conditions were used to represent bulk phase conditions. Energy minimization was then performed using the steepest decent algorithm at a step size of 0.01 nm and 1000 kj/mol/nm tolerance. The system was then equilibrated for 1 ns with, V-rescale temperature coupling (0.1 ps tau) and semi-isotropic Berendsen pressure coupling (1 ps tau)[53,54]. Electrostatic interactions were treated with the Particle Mesh Ewald (PME) algorithm and a cubic spline interpolation of 4 with 0.1 grid spacing for the fast Fourier transform[55]. Isothermal compressibility was set to 4.5e$^{-5}$ bar$^{-1}$. Temperature was set to 298 K. Hydrogen atoms were constrained with LINCS holonomic constraints[56]. The time step used was 2 fs with heavy atom restrains of 1000 kJ/mol/nm². Production runs of each equilibrated system was then initiated for 100 ns, with Cα atom restrains of 1000 kj/mol/nm². Nosé-Hoover was used for temperature and pressure coupling with respective coupling constants of 0.5 and 1 ps at 298 K[55,57,58]. Analysis of RMSDs were done with in-house python scripts and used MDAnalysis v2.0.0[59,60]. NTH containing hCALHM1Δct with POPC, cholesterol and apo-state systems were simulated for 100 ns per repeat with 8, 9 and 9 independent repeats per system respectively.

## Reporting summary

Further information on research design is available in the Nature Portfolio Reporting Summary linked to this article.

# Data availability

Previously published cryo-EM models and micrographs used in this study include PBD-6VAM and EMPIAR #10444. The cryo-EM maps of chCALHM1Δct, hCALHM1Δct, RuR-hCALHM1$_{I109W}$Δct (c8), RuR-hCALHM1$_{I109W}$Δct (c1), and hCALHM1$_{I109W}$Δct generated during this study have been deposited in the Electron Microscopy Data Bank with accession codes: EMD-40230, EMD-40231, EMD-40232, EMD-40233, EMD-40229. Models have been deposited into the Protein Data Bank with accession codes: 8GMQ, 8GMR, 8S8Z, 8S90, 8GMP. The MD simulations data have been deposited to Zenodo (https://zenodo.org/deposit/7706705#). Source data are provided with this paper.

# Code availability

The MD simulations codes have been deposited to github (https://github.com/Maxim-93/CALHM1).

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

## Acknowledgements
We would like to thank D. Thomas and M. Wang for managing the cryo-EM facility and the computing facility at Cold Spring Harbor Laboratory, respectively. We would also like to thank Dr. R. Corey and Dr. T. Piggot for helpful discussions regarding the coarse-grained MD simulations. This work was supported by NIH NS111745, NS113632 and MH085926, Austin's purpose, Robertson funds at Cold Spring Harbor Laboratory, Doug Fox Alzheimer's fund, Heartfelt Wing Alzheimer's fund, and the Gertrude and Louis Feil Family Trust (to H.F.) and the Charles H. Revson Senior Fellowship in Biomedical Science (to J.L.S.).

## Author contributions
J.L.S. conducted cryo-EM experiments and electrophysiology. M.E. conducted MD simulations. R.G. conducted electrophysiology. J.L.S., M.E., R.G. and H.F. wrote the manuscript.

## Competing interests
The authors declare no competing interests.
