## [Peer Review File · Nature Communications]

Structure of human CALHM1 reveals key locations for channel regulation and blockade by ruthenium redREVIEWER COMMENTS

Reviewer #1 (Remarks to the Author):

Comments to the authors:

This manuscript from Syrjanen et al. reveals the first structure of the human CALHM1 channel, which is the most studied CALHM isoform. Although non-mammalian CALHM1 structures were previously resolved, this work confirms that CALHM-1 channel architecture is well conserved among species and addresses two relevant questions: the role of conserved hydrophobic domains on channel function and the mechanism of blockade by Ruthenium red. Overall, this is a very good paper that takes advantage of complimentary methodologies, including molecular dynamics and electrophysiology, to strengthen their conclusions. While the structural data is solid, the role of lipid binding in conserved hydrophobic pockets and subsequent regulation of channel function may require a better demonstration to support author's conclusions. See specific concerns below:

Major concerns

1. Molecular dynamics (MD) suggest that the binding of POPC is rapid and energetically more favorable than cholesterol, and cholesterol (in a POPC environment) destabilizes CALHM-1, indicating that cholesterol might compete and bind to the hydrophobic pocket despite the favorable conditions for POPC binding. This finding is very interesting and could support the proposed functional relevance of the conserved hydrophobic in CALHM channels. I believe it is feasible that the authors complement their MD data with simple functional experiments interrogating whether cholesterol-enriched environments affect CALHM-1 channel function.
2. Some mutations from extended Figure 5 do not displayed changes in the density current plotted in Fig. 3B, see for example A116W mutant. In my view, it is hard to elaborate reliable conclusions for a role of a lipid pocket on channel gating/function from single channel mutagenesis and analysis of the maximal current recorded at +100mV pulse. The authors should consider a more exhaustive analysis of electrophysiological experiments (calcium or voltage-dependence and activation/deactivation kinetics) that might offer more insights into the role of the hydrophobic pocket on channel function. Could the authors use MD to support that some mutations critically affect or not POPC binding? This would strengthen their conclusions.
3. Regarding to the ab-FSEC assay to measure the expression of proteins in the plasma membrane, it is not clear if these values were normalized to total protein and what is the n value for these experiments. Also, quantification and statistical analysis are needed for proper data analysis. Secondly, if channel expression is variable between groups, as suggested by the ab-FSEC experiments shown in Fig. 3D, then, data shown in Figure 3B could be being misinterpreted. For example, I109W shows less expression than wild type but higher current density. Conversely, A199W shows higher expression but similar current

density than wild type, suggesting the opposite effect on gating/conductance that I109W. Based on this discrepancy, it is difficult to interpret the effect of destabilization of the hydrophobic pocket on channel function. If channel expression is effectively higher for I109W, then, how can the authors explain that destabilization of the hydrophobic pocket by replacing hydrophobic residues with tryptophan has opposite effects on channel function? Based on the current data, the effect of mutations on gating properties could not be caused because of affecting lipid binding/interaction, but just because of the mutation per se. Authors should address this issue with additional experiments. Finally, the authors did not explore what happened with the plasma membrane expression of the other 4 mutants in the hydrophobic pocket, so a proper interpretation of how mutants affect channel function cannot be made based on current data.

4. From the structure obtained in the presence of RuR, authors state that “Ile109Trp ‘caps’ the hydrophobic pocket to favor stable lipid binding” (Line 206). The current manuscript does not contain enough data supporting conclusions about lipid binding stability. Authors should confirm by modeling that this mutation favor lipid binding. Also, if MD simulations suggests that lipid binding is fast and energetically favorable, it is not clear how an enhanced lipid binding in an environment saturated with lipids (i.e., lipid bilayer) could regulate the CALHM1 channel function as proposed for I109W. With the current structural and functional data, authors cannot exclude that increased activity of CALHM1 I109W are caused by the mutation per se and not because of lipid binding or rearrangements of lipids in the modified hydrophobic pocket.

5. The lack of a figure showing the direct comparison between structures of CALHM1 I109W in absence and presence of RuR makes difficult to understand the similarity and major differences between both structures. Does the presence of RuR affect the overall rearrangement of CALHM1 domains? Also, authors state that “the more ordered TMD1 facilitates the interaction between the lipid and Phe19 located between TMD1 and NTH to order and position the pore-forming NTH in the ‘upright’ manner...” (Lines 208-210). Since a direct comparison between mutant and wild type channel is not shown and critical residues in the TMD1-NTH domains were not resolved for the wild type channels, the author’s statement is not straightforward when reading the manuscript. Can the proposed rearrangement be caused by RuR or is it caused by the mutation I109W itself? Additionally, considering that this is the first human CALHM-1 structure and NTH which lines the pore o CALHM1 was resolved in the presence of RuR, authors should consider incorporating a graph with the calculated radius along the pore, and compare it with the other CALHMs structures.

Minor concerns:

1. Please incorporate in method section the details of reconstitution of human CALHM-1, and hCALHM-1-Ile109Trp into nanodiscs. Currently, data refers only to chicken CALHM-1
2. Why Cryo-EM density of TMD1 was omitted for chicken and human CALHM-1 structures in Extended data 1D and 2D? It is not clear from the manuscript the unresolved residues in these structures, particularly those in NTH and TMD1.
3. Page 19, Line 414. There is a repeated phrase.

4. Extended Data Table 3. It is not clear why authors perform unpaired and paired t-test using the same datasets. Also, legend states that statistics comes from Figure 5b-c but it seems that refers to Figure 5D.
5. In the manuscript, mutation Pro86Leu mutation is stated as a human mutation but site 86 in human CALHM-1 is not a Proline but a Leucine. Authors state that Pro86 was not resolved (Line 105) and they modeled it using the chicken CALHM-1 structure (Extended Data Figure 3E). Please clarify.
6. Figure 2B. I assume that colored beads represent lipids (POPC and/or cholesterol) Please clarify this in the manuscript and indicate what represent each color.
7. Line 169-170. Authors state that reduced current by V192T mutation is likely due to destabilization of the hydrophobic pocket caused by steric clash, but mutations in residues A116 and A11W currents are not affected even when these residues are closer and more likely for steric clash.
8. From CG-MD simulations, it is not clear that lipids occupy the eight hydrophobic pockets, as authors state. Can the authors incorporate a supporting file (e.g., a movie) to support this?
9. Fig 5B, three points are not enough to fit data using Boltzmann equation. More points are needed to get a reliable fitting.
10. Current traces of blockade by RuR in Q10R I109W group are not representative of what authors show in final data quantification. Please replace it by better representative traces.
11. Pore diameter is shown slightly higher in I109W vs WT (~41 vs 36 Å, respectively). These values are confusing since it is not clear if these distances were calculated from the same C α . Please clarify in the manuscript the details for measuring these distances.
12. Method section indicates that internal pipette solution for electrophysiological recordings is composed by 147mM NaCl. Is this correct?
13. Line 465, there is a spelling mistake here.
14. Line 460. Please specify what experiments were performed with this solution.

Reviewer #2 (Remarks to the Author):

Manuscript ID: NCOMMS-22-41234

Structure of human CALHM1 reveals key locations for channel regulation and blockade by ruthenium red
Syrjänen et al.

CALHM1 forms a large pore channel that functions as an ATP release channel in a voltage-dependent manner. This channel is associated with neurotransmitter release in taste bud cells and is also suggested to be a cause of Alzheimer's disease. The regulation of this channel would contribute to treating these events and human symptoms. However, there are many unresolved issues about this channel. For instance, the previous structural studies have shown the different oligomeric numbers of CALHM1 channels from different species while the determinant of oligomeric assembly remains unclear. It has been presented that RuR becomes a blocker of the CALHM1 channel, but the blocking mechanisms and binding site of RuR are still provisional and not clearly understood. The structural basis regarding the gating regulation of CALHM1 remains mysterious. These are partially due to the unresolved N-terminal domain due to the flexibility of CALHM1, which is supposed to exist possibly in the pore.

In this manuscript, Syrjänen et al. have reported the cryo-EM structures of human and chicken CALHM1 (hCALHM1, chCALHM1) for comparison. They solve the mutant hCALHM structures where RuR is bound or non-bound and present that the density in the middle of the pore surrounded by the pore lining N-terminal helices (NTH) is assigned to be RuR. Because of a physical block on the pore pathway, this structure is supposed to be closed or non-permeable. MD simulation is accompanied to show that the hydrophobic pocket prefers a phospholipid rather than cholesterol, and this lipid binding facilitates the stability of the CALHM1 channels.

In general, the structural determination is reliable as there are few unreasonable parts in image analysis and modeling. The combination of patch-clamp recordings and MD simulation along with cryo-EM is a preferable strategy for understanding the structural basis of membrane channel proteins. While the visualized NTH and RuR in CALHM1 are interesting, I should admit that there are concerns regarding the authors' research methods, interpretation, and novelty of the structures. The authors should address the following issues I raised to make this manuscript more persuasive and comprehensible.

Major concerns:

1.

It is described that the ordered NTHs in an upright manner would maximize the pore size and ion permeability. However, NTH is visible only when RuR is added to the I109W mutant. Isn't this a blocked structure without any activity? This structure does not guarantee that the upright NTHs make an open state unless the structure with NTHs surrounding the pore without RuR is determined. Because the RuR-free structure of hCALHM1I109W Δ ct does not show NTH, this assertion is not reasonable. In addition, the RuR bound hCALHM1 has the mutation of I109W, not wild type (WT). The authors show that I109W upregulates the channel activity (Fig. 3b), but it could be an artificial effect by the mutation, and the same is true for the RuR blockade because the structure of hCALHM1 Δ ct shows no density for NTH nor RuR. Given that, it is unclear what foundation Fig. 6 is drawn. The Fig.6 right panel stating "RuR plugging

open channel” is wrong as a claim, and this is for Δ ct and I109W, not WT. The left of Fig. 6 is even more unsubstantiated.

2.

The authors assigned RuR on the middle pore density. However, there is an ambiguity regarding the densities for RuR and NTHs in the current cryo-EM map, which does not reveal the feature of side chains of NTH nor make sure of the RuR orientation. The foundation of this assignment is that the oblong pore density corresponds to the size of RuR, and it disappeared in the RuR-free structure of hCALHM1I109W Δ ct (Line 219). In the RuR-free structure of hCALHM1I109W Δ ct, the densities assumed to be NTH along with RuR are lost. This is not enough to assign the orientation of RuR or the interacting side chains of hCALHM1I109W Δ ct with RuR. What if an unresolved peptide portion of hCALHM1I109W Δ ct perhaps contributes to these densities? The decreased inhibition by Q10R109W and Q13R109W (Fig. 5c, d) does not necessarily prove the repulsive effect against RuR nor that those two residues have direct interactions with RuR to fix RuR in the middle of the pore. Indeed, the partial inhibition of activity is still observed in those mutants (Fig. 5d). It is also not convincing that the concentration of RuR differs between 20 μ M in Fig. 5d and 50 μ M for structure determination. The assignment of RuR based on indirect evidence may cause more confusion.

3.

Related to the above, the authors use the C-terminal deletion mutant (Δ ct). In Ext. Data Fig. 5, no functional data for the full-length WT is shown, and hCALHM1 Δ ct is deemed WT, which is confusing. In this manuscript, no data for the full-length WT is shown, and all interpretations are based on hCALHM1 Δ ct. How can we be sure that this is physiologically significant, not an artificial effect? The authors presented that the C-terminal portion may facilitate an octameric assembly, but 9-mer itself is the consequence of Δ ct, which does not happen to the native CALHM1. Has the importance of the deleted C-terminus in determining oligomeric number been generally shown for other CALHMs subtypes, and how biologically significant is it?

4.

The authors demonstrated in MD simulation that the phospholipid binding in this hydrophobic pocket stabilizes the CALHM1 channel structure. Have the authors examined the calculation starting from the model without lipids in the hydrophobic pocket? How unstable the CALHM1 channel is when nothing occupies the hydrophobic pocket?

5. Lipid binding hydrophobic pocket.

a) Line 189, "On the other hand,.... the Val192Trp mutant likely stems from protein instability and a lack of trafficking to the plasma membrane."

Why do the authors say that only V192W lacks stability and membrane trafficking? In Fig. 3d, the signals for I109W and A199W are also weaker than for WT. It is likely that the mutants of I109W and A199W somehow lose the protein stability or normal trafficking if this analysis ensures quantitativity. Furthermore, only three mutants, not all studied, are shown in Fig. 3b. This representation is biased.

Another confusing point in Fig. 3 is that the method to investigate surface expression is indirect. If possible, fluorescence microscopy is a more straightforward way to understand membrane transport. Is there any reason not to do so?

b) Fig. 3

The authors focused on 7 mutants around the hydrophobic pocket and generated Trp substitution mutants. The interpretation of Fig. 3b (Line 167, "First, incorporating the Val192Trp mutation.....") is incomprehensible. Looking at the structure model, the mutants of V112W, A116W, and A199W can cause the steric hindrance against lipids as interpreted for V192W, but it is noted that these tryptophan mutants are not positioned to strengthen or weaken hydrophobic contacts in the pocket. Why is it considered that only V192W causes pocket destabilization and the other three mutants do not?

Finally, are these residues related to Alzheimer's disease?

c) Line 205,

"Our structural analysis revealed....Ile109Trp 'cap' the hydrophobic pocket..."

The lipid density is also found in the structure of hCALHM1 Δ ct, suggesting that the I109 cap is not necessary for lipid binding. This should be addressed.

Minor:

The authors have reported the structure of chCALHM1 in nanodiscs in 2020 (Syrjänen et al. NSMB (2020)). What is the novelty in the RuR-free CALHM1 structures in this manuscript? Is there a significant difference? This should be addressed.

Fig. 3b

The error bars for I109W and T196W are pretty long. Is there any reason for the large variability?

For CALHM2, a different binding site of RuR has been shown. Is there any relevance to this study?

Reviewer #3 (Remarks to the Author):

This manuscript describes the structure of human CALHM1. This group reported the first cryo-EM structures of CALHM channels a couple of years ago, including chicken CALHM1. Novel observations here: identification of phospholipid binding in the structure (albeit in the same site previously identified by others as occupied by hemi-cholesterol; and observed also previously in other CALHM structures) with MD simulations suggesting that it helps to stabilize the channel structure. Second, observation of the channel inhibitor ruthenium red (RuR) in the channel pore, although another group previously demonstrated such a structure, albeit in a different CALHM channel paralog and with binding in a different site.

The work is well done, but fundamental new insights into the structural bases for CALHM channel permeation and gating CALHM channel gating have not been provided that go beyond what these and other groups have previously reported. In particular, the orientation of the Nths with relationship to gating and permeation is still speculated upon, as it has been by other groups, without new hard data to support the model in Fig 6. My comments below touch on some of these aspects.

1. The authors suggest that the I109W mutation stabilized the Nth in the upright position, but then suggest that RuR is responsible for this structural placement. Furthermore, Nth seems not to be resolved in such a position in the RuR-free structure (Ext Fig 10). The authors need to clarify that they are speculating in either case. It could be that it is the RuR that stabilizes the structure and/or the mutation that does that. The former seems more likely.

2. The authors demonstrate that RuR inhibits channel currents, and they provide a structure of a mutant channel in the presence of RuR. A couple of things should be addressed by the authors. First, why did the authors not solve the structure of WT CALHM in the presence of RuR...this might also help to address the previous comment. Second, the authors need to comment on the environment surrounding the RuR in the pore. Does water surround the RuR? Does RuR manage in this structure to sterically block a 16Å diameter pore to small ions? Intuitively, this seems unlikely, but what does the structure say? How

do the authors know that THIS structure is one of a “blocked” channel? Another study demonstrated RuR binding to a different site in the channel (albeit a different CALHM). Notably, the Q10 and Q13 R mutations did not abolish RuR inhibition. Of note, the inhibition in the example trace for Q10R is inconsistent with the dot plot summary; and the currents for the Q16R channel in the absence of RuR seem different from WT channel behavior. Are the current densities for the Q-to-R mutants different from WT in the absence of RuR?

3. Please clarify how structures of I109W-CALHM1 were obtained in the absence of RuR. The text states that it was not possible because of toxicity, yet data are shown and referred to.

4. The currents in Ext Fig 5 should be presented as current densities.

5. Regarding the I109W effects on currents. It's not obvious from the description of the structure why only this mutation enhanced the currents. It is surprising that the authors did not extend their MD simulations to this mutant to confirm structural stabilization. In addition, how can the authors know that a “capping and stabilizing” effect accounts for the enhanced current density?

Reviewer #4 (Remarks to the Author):

The manuscript reports on a structural investigation of the voltage-dependent channel CALHM1. cryoEM, electrophysiology, and molecular dynamics simulations are used to clarify some aspects of channel activity and its modulation, the structure of the pore and the binding site of ruthenium red, known to inhibit the channel. The results are original and, given the mechanistic insight they provide, of interest to a wide community of scientists. The conclusions are in general supported by the data. In particular, the molecular simulations are competently performed and properly analyzed: the arguments presented in support of the hypothesis that the pocket prefers POPC over cholesterol are compelling. Overall, I am enthusiastic about the paper and I recommend publication. There are only few clarifications and comments that I recommend:

1) I am a bit unclear as to whether or not the NTH has been resolved in all the structures or only in the RuR-bound one (my understanding is that the latter is true). I suspect that the information is reported somewhere, but I suggest the authors to specify more clearly which residues were modeled in each structure.

2) Related to the first question: what model was used for the MD simulations? Was the NTH present? How about RuR, was it modeled?

3) If NTH was included in the simulation, was any insight gathered about the dynamics of this helix in presence and in absence of RuR? Do these observations support the proposed mechanism shown in Fig 6?

4) A crucial question regarding the lipid binding pocket is the specificity for particular lipid species. For instance, the channel might bind selectively unsaturated or saturated lipids. Which one of the two lipid tails (palmitic or oleic) is in contact with TMD1 and NTH? Did the authors simulate alternative binding poses with one or the other lipid tail in contact with NTH?

Responses to the reviewers:

We thank the reviewers and editors for comments and suggestions. We revised the manuscript according to the reviewers' comments. Responses to comments by the reviewers are in light blue.

Reviewer #1 (Remarks to the Author):

Comments to the authors:

This manuscript from Syrjanen et al. reveals the first structure of the human CALHM1 channel, which is the most studied CALHM isoform. Although non-mammalian CALHM1 structures were previously resolved, this work confirms that CALHM-1 channel architecture is well conserved among species and addresses two relevant questions: the role of conserved hydrophobic domains on channel function and the mechanism of blockade by Ruthenium red. Overall, this is a very good paper that takes advantage of complimentary methodologies, including molecular dynamics and electrophysiology, to strengthen their conclusions. While the structural data is solid, the role of lipid binding in conserved hydrophobic pockets and subsequent regulation of channel function may require a better demonstration to support author's conclusions. See specific concerns below:

Major concerns

1. *Molecular dynamics (MD) suggest that the binding of POPC is rapid and energetically more favorable than cholesterol, and cholesterol (in a POPC environment) destabilizes CALHM-1, indicating that cholesterol might compete and bind to the hydrophobic pocket despite the favorable conditions for POPC binding. This finding is very interesting and could support the proposed functional relevance of the conserved hydrophobic in CALHM channels. I believe it is feasible that the authors complement their MD data with simple functional experiments interrogating whether cholesterol-enriched environments affect CALHM-1 channel function.*

We appreciate the reviewer's suggestion to experimentally explore the effect of cholesterol on channel activity. As the mammalian cell membrane is naturally very cholesterol-rich, we compared ion channel activities of CALHM1 in mammalian cells to cells treated with a cholesterol-depleting agent. To do so, we performed whole cell patch clamp on human CALHM1 Δ ct heterologously expressed in HEK293 cells in both cholesterol-rich (0 mM methyl- β -cyclodextrin) and cholesterol-depleted conditions. For cholesterol depletion, cells were incubated with 10 mM methyl- β -cyclodextrin prior to recordings. Comparing the current densities at 100 mV revealed upregulation of CALHM1 channel activity when membrane cholesterol is reduced (see figure on the left).

This electrophysiological result is in agreement with our atomistic MD simulations observations, showing an increased variance of RMSD values in cholesterol bound structures compared to that of POPC-bound structures. More detailed experiments may be required to conclude this speculation in the future. For this work, we would like to show reviewers this important electrophysiological result here.

2. Some mutations from extended Figure 5 do not displayed changes in the density current plotted in Fig. 3B, see for example A116W mutant. In my view, it is hard to elaborate reliable conclusions for a role of a lipid pocket on channel gating/function from single channel mutagenesis and analysis of the maximal current recorded at +100mV pulse. The authors should consider a more exhaustive analysis of electrophysiological experiments (calcium or voltage-dependence and activation/deactivation kinetics) that might offer more insights into the role of the hydrophobic pocket on channel function. Could the authors use MD to support that some mutations critically affect or not POPC binding? This would strengthen their conclusions.

We agree that MD simulations on the upregulating Ile109Trp mutation would strengthen the conclusions of the manuscript. We have now performed PMF calculations on the human Ile109Trp CALHM1 Δ ct mutation. These data indicate that the binding of the phospholipid POPC to the hydrophobic pocket in Ile109Trp CALHM1 Δ ct is more energetically favorable than the binding in human CALHM1 Δ ct (-9.2 kcal/mol in Ile109Trp vs -5.5 kcal/mol in wild-type). Cholesterol binds more favorably to human Ile109Trp CALHM1 Δ ct compared to human CALHM1 Δ ct (-1.45 kcal/mol in Ile109Trp vs +5.5 kcal/mol in wild-type), but less favorably than POPC, indicating that the Ile109Trp mutation has not affected lipid specificity. This new result is incorporated into the revised Figure 4d.

3. Regarding to the *ab*-FSEC assay to measure the expression of proteins in the plasma membrane, it is not clear if these values were normalized to total protein and what is the *n* value for these experiments. Also, quantification and statistical analysis are needed for proper data analysis. Secondly, if channel expression is variable between groups, as suggested by the *ab*-FSEC experiments shown in Fig. 3D, then, data shown in Figure 3B could be being misinterpreted. For example, I109W shows less expression than wild type but higher current density. Conversely, A199W shows higher expression but similar current density than wild type, suggesting the opposite effect on gating/conductance that I109W. Based on this discrepancy, it is difficult to interpret the effect of destabilization of the hydrophobic pocket on channel function. If channel expression is effectively higher for I109W, then, how can the authors explain that destabilization of the hydrophobic pocket by replacing hydrophobic residues with tryptophan has opposite effects on channel function? Based on the current data, the effect of mutations on gating properties could not be caused because of affecting lipid binding/interaction, but just because of the mutation *per se*. Authors should address this issue with additional experiments. Finally, the authors did not explore what happened with the plasma membrane expression of the other 4 mutants in the hydrophobic pocket, so a proper interpretation of how mutants affect channel function cannot be made based on current data.

The goal of our original analysis was to evaluate the surface expression and assembly of mutations that resulted in statistically significant upregulated or downregulated ion channel activities as measured by current density at 100 mV. Nevertheless, we agree with the reviewer's suggestion to extend our surface expression measurements to all the mutants analyzed in Figure 3. We performed the surface expression assay as outlined in the revised Methods section. We repeated this assay to produce between 2 (Leu67Trp, Thr196Trp) to 3 (Ile109Trp, Ala199Trp, Val192Trp, Ala116Trp, Val112Trp and hCALHM1 Δ ct) biological replicates and analyzed the surface expression by western blot in addition to our earlier FSEC analysis. We probed for the CALHM1 expression and used a β -actin antibody for normalization. These data have been added to the revised Figure 3d. To summarize our findings, Ala199Trp hCALHM1 Δ ct, hCALHM1 Δ ct and Ile109Trp hCALHM1 Δ ct at express at higher levels

compared to the other mutants. Indeed, consistent with our FSEC analysis, Ala199Trp hCALHM1 Δ ct has the highest expression of all the mutants. The other mutants Leu67Trp, Val112Trp, Ala116Trp, Thr196Trp, and Val192Trp are present at drastically lower levels on the cell surface.

These data indicate that the location of the mutation within the hydrophobic pocket affects cell membrane expression of hCALHM1 Δ ct. Depending on the position, expression may either be increased or decreased compared to hCALHM1 Δ ct. Taken together with our electrophysiology studies, these data suggest that mutations of different positions within the hydrophobic pocket have profoundly differing effects on channel activity. Ala199Trp hCALHM1 Δ ct appears to have reduced ion channel activity compared to hCALHM1 Δ ct, as the current densities at 100mV are similar but the surface expression of Ala199Trp hCALHM1 Δ ct is higher. It is possible that the activities of Leu67Trp, Val112Trp, Ala116Trp, and Thr196Trp hCALHM1 Δ ct may be upregulated compared to hCALHM1 Δ ct, as their surface expression is multiple-fold lower.

4. From the structure obtained in the presence of RuR, authors state that “Ile109Trp ‘caps’ the hydrophobic pocket to favor stable lipid binding” (Line 206). The current manuscript does not contain enough data supporting conclusions about lipid binding stability. Authors should confirm by modeling that this mutation favor lipid binding. Also, if MD simulations suggests that lipid binding is fast and energetically favorable, it is not clear how an enhanced lipid binding in an environment saturated with lipids (i.e., lipid bilayer) could regulate the CALHM1 channel function as proposed for I109W. With the current structural and functional data, authors cannot exclude that increased activity of CALHM1 I109W are caused by the mutation per se and not because of lipid binding or rearrangements of lipids in the modified hydrophobic pocket.

The reviewer is correct about the potentiation effect of I109W caused by the mutation itself. The take home message is that the phospholipid is an integral part of structure that interacts with TMD, which in turn promotes interaction between TMD1 and NTH to stabilize the pore. Here are the experimental supports.

- 1) The capping effect of Ile109Trp is observed in both RuR bound and unbound cryo-EM structures. The cryo-EM structures on I109W showed that there is no unexpected ‘artifactual’ changes from the WT.
- 2) Ile1109Trp has more ordered TMD1 density in our cryo-EM structure. Together with the more resolved RuR-bound structure that shows TMD1-NTH interactions, allow us to conclude that there is a network between the lipid binding – TMD – NTH.
- 3) To further validate this network and the effect of I109W, we conducted a new set of MD simulations on I109W in addition to the mutant (shown in the revised Figure 4d). The PMF calculation clearly indicated that POPC binds more favorably to the pocket in Ile109Trp than WT.
- 4) Our original and new sets of MD simulations show that POPC binding is energetically more favorable than cholesterol binding to the pocket. This indicates a possibility that cholesterol content could affect the channel function although it remains to be seen if this happens in more sophisticated biological system (e.g., neurons vs. peripheral tissue cells with different content of cholesterol). Our data on beta-cyclodextrin (above) preliminarily shows that such regulation may occur.

The above experimental results are more clearly addressed in the revised manuscript.

5. *The lack of a figure showing the direct comparison between structures of CALHM1 I109W in absence and presence of RuR makes difficult to understand the similarity and major differences between both structures. Does the presence of RuR affect the overall rearrangement of CALHM1 domains? Also, authors state that “the more ordered TMD1 facilitates the interaction between the lipid and Phe19 located between TMD1 and NTH to order and position the pore-forming NTH in the ‘upright’ manner...” (Lines 208-210). Since a direct comparison between mutant and wild type channel is not shown and critical residues in the TMD1-NTH domains were not resolved for the wild type channels, the author’s statement is not straightforward when reading the manuscript. Can the proposed rearrangement be caused by RuR or is it caused by the mutation I109W itself? Additionally, considering that this is the first human CALHM-1 structure and NTH which lines the pore of CALHM1 was resolved in the presence of RuR, authors should consider incorporating a graph with the calculated radius along the pore, and compare it with the other CALHMs structures.*

As described on lines 258, the structures of hCALHM1I109W Δ ct in the presence and absence of RuR are similar in TMD2-4 and CTH (RMSD = 0.541 Å; over 197 C α positions), indicating that the binding of RuR only affects the structure of the central pore comprised of TMD1 and NTHs. It does not cause an overall structural rearrangement of these CALHM1 domains.

The NTHs are resolved only when RuR is in complex with human Ile109Trp CALHM1 Δ ct. The challenge in resolving the NTHs is most likely due to the highly mobile nature of the NTHs, suggested in the recent MD simulations study (Ren et al., 2022).

We illustrate the effect of the NTHs on the pore diameter in Supplementary Figure 11 and have incorporated a graph outlining this effect in Supplementary Figure 11c.

Thanks for the suggestion.

Minor concerns:

1. *Please incorporate in method section the details of reconstitution of human CALHM-1, and hCALHM-1-Ile109Trp into nanodiscs. Currently, data refers only to chicken CALHM-1*

The Methods section has been updated to include references to all proteins studied by cryo-EM. In brief, the protocol for reconstitution into nanodiscs was similar for all constructs (chicken CALHM1, human CALHM1, human CALHM1 Ile109Trp, human CALHM1 Ile109Trp-RuR), with the difference being that hCALHM-1-Ile109Trp-RuR was in a RuR-containing buffer.

2. *Why Cryo-EM density of TMD1 was omitted for chicken and human CALHM-1 structures in Extended data 1D and 2D? It is not clear from the manuscript the unresolved residues in these structures, particularly those in NTH and TMD1.*

We now show cryo-EM density for TMD1 for the chicken and human calhm1 Δ ct structures (Supplementary Fig. 1d and 2d). We also summarize the most N-terminal residues modeled in each structure here:

Chicken calhm1 Δ ct: Ala27

human calhm1 Δ ct: Ala30

human calhm1 Δ ct Ile109Trp with RuR: Met1

human calhm1 Δ ct Ile109Trp without RuR: Gly25

3. Page 19, Line 414. There is a repeated phrase.

We have corrected this.

4. Extended Data Table 3. It is not clear why authors perform unpaired and paired t-test using the same datasets. Also, legend states that statistics comes from Figure 5b-c but it seems that refers to Figure 5D.

The datasets are used to compare 1) the same cell before and after ruthenium red (RuR) application and 2) the extent of RuR inhibition with different point mutations.

Paired t-tests were used to compare the same cell before and after RuR application for all the different constructs studied (i.e. Ile109Trp hcalhm1 Δ ct control vs. Ile109Trp hcalhm1 Δ ct + RuR; Gln10Arg/Ile109Trp hcalhm1 Δ ct control vs. Gln10Arg/Ile109Trp hcalhm1 Δ ct + RuR, etc.).

Unpaired t-tests with Welch's correction were used to compare the extent of RuR inhibition between the arginine mutants and the channel in the Ile109Trp hcalhm1 Δ ct background (Ile109Trp hcalhm1 Δ ct + RuR vs. Gln10Arg/Ile109Trp hcalhm1 Δ ct; Ile109Trp hcalhm1 Δ ct in the presence of RuR vs. Gln13Arg/Ile109Trp hcalhm1 Δ ct in the presence of RuR, etc.). This is stated in the Methods section and in Supplementary Table 4.

The reviewer is correct; the statistics refer to the original Figure 5D. The figure legend has been modified accordingly (corresponding now to Figure 6d and Supplementary Table 4).

5. In the manuscript, mutation Pro86Leu mutation is stated as a human mutation but site 86 in human CALHM-1 is not a Proline but a Leucine. Authors state that Pro86 was not resolved (Line 105) and they modeled it using the chicken CALHM-1 structure (Extended Data Figure 3E). Please clarify.

As described by the first paper that discovered the CALHM1 gene and protein (Dreses-Werringloer et al., 2008), amino acid position 86 is affected by a single nucleotide polymorphism in some population groups that results in a leucine at site 86 rather than a proline. In humans, amino acid 86 may therefore be a proline or a leucine. In our chicken CALHM1 structure, we were able to model the cytoplasmic loop corresponding to chicken residues 85-90. In amino acid sequence alignments of human and chicken CALHM1, position 86 in human (Pro86 or Leu86) aligns to position 85 (Gln85) in chicken.

6. Figure 2B. I assume that colored beads represent lipids (POPC and/or cholesterol) Please clarify this in the manuscript and indicate what represent each color.

Thank you for pointing this out. We will amend the manuscript accordingly.

7. Line 169-170. Authors state that reduced current by V192T mutation is likely due to destabilization of the hydrophobic pocket caused by steric clash, but mutations in residues A116 and A11W currents are

not affected even when these residues are closer and more likely for steric clash.

Val192Trp hCALHM1 Δ ct is the only mutation to show a statistically significant reduced current density compared to hCALHM1 Δ ct. This is why we focused on determining an explanation for the reduced current density we observed and evaluated the cell membrane expression and assembly of this mutant.

As explained in our response to Major Concern #3 above, we have now extended our surface expression measurements to all the hydrophobic pocket mutants analyzed in Figure 3. The surface expression of Leu67Trp, Val112Trp, Ala116Trp, and Thr196Trp hCALHM1 Δ ct are all multiple-fold lower than that of hCALHM1 Δ ct. This is consistent with a destabilizing effect on structural stability and/or defect in plasma membrane trafficking.

8. From CG-MD simulations, it is not clear that lipids occupy the eight hydrophobic pockets, as authors state. Can the authors incorporate a supporting file (e.g., a movie) to support this?

We have incorporated representative snapshots from three independent unbiased CG-MD simulations into Supplementary Figure 12b to help demonstrate the eight lipid binding positions more clearly. Furthermore, these CG-MD simulations have now been performed on an hCALHM1 Δ ct model containing the NTHs.

9. Fig 5B, three points are not enough to fit data using Boltzmann equation. More points are needed to get a reliable fitting.

We agree with the reviewer's comment and are presenting the data as a bar chart in the revised Figure 6b.

10. Current traces of blockade by RuR in Q10R I109W group are not representative of what authors show in final data quantification. Please replace it by better representative traces.

This trace has been replaced. A better representative Q10R_{I109W} trace is shown in the revised Figure 6.

11. Pore diameter is shown slightly higher in I109W vs WT (~41 vs 36 Å, respectively). These values are confusing since it is not clear if these distances were calculated from the same C α . Please clarify in the manuscript the details for measuring these distances.

We have clarified which C α has been used to measure the pore diameter. Using the C α of Gln33 for Ile109Trp hCALHM1 Δ ct gives a pore diameter of ~40 Å and using the same C α for hCALHM1 Δ ct gives a pore diameter of ~41 Å. The figures and figure legends (of Figure 1; Supplementary Figures 3 and 10) are modified accordingly to more clearly reflect this.

12. Method section indicates that internal pipette solution for electrophysiological recordings is composed by 147mM NaCl. Is this correct?

The whole cell patch clamp experiments measuring channel activity of hCALHM1 Δ ct and the hydrophobic pocket mutants Ala116Trp, Ala199Trp, Leu67Trp, Ile109Trp, Thr196Trp, Val112Trp, and

Val192Trp hCALHM1 Δ ct were done using an internal pipette solution containing 147 mM NaCl, 10 mM HEPES, 10 mM EGTA; pH 7.0 with NaOH.

Electrophysiology recordings interrogating the effect of ruthenium red (RuR) on Ile109Trp hCALHM1 Δ ct were performed with a pipette solution of 110 mM Cs-gluconate, 30 mM CsCl, 5 mM HEPES, 5 mM BAPTA, 4 mM NaCl, 2 mM MgCl₂, 0.5 CaCl₂, 2 mM ATP-Na, 0.3 mM GTP-Na, pH 7.35 with CsOH.

13. Line 465, there is a spelling mistake here.

This has been corrected in the text.

14. Line 460. Please specify what experiments were performed with this solution.

This has been clarified in the text. The whole cell patch clamp experiments measuring channel activity of Ala116Trp, Ala199Trp, Leu67Trp, Ile109Trp, Thr196Trp, Val112Trp, and Val192Trp hCALHM1 Δ ct and hCALHM1 Δ ct were performed in a bath solution of 147 mM NaCl, 13 mM glucose, 2 mM KCl, 2 mM CaCl₂, 1 mM MgCl₂, 10 mM HEPES-NaOH pH 7.3.

Reviewer #2 (Remarks to the Author):

CALHM1 forms a large pore channel that functions as an ATP release channel in a voltage-dependent manner. This channel is associated with neurotransmitter release in taste bud cells and is also suggested to be a cause of Alzheimer's disease. The regulation of this channel would contribute to treating these events and human symptoms. However, there are many unresolved issues about this channel. For instance, the previous structural studies have shown the different oligomeric numbers of CALHM1 channels from different species while the determinant of oligomeric assembly remains unclear. It has been presented that RuR becomes a blocker of the CALHM1 channel, but the blocking mechanisms and binding site of RuR are still provisional and not clearly understood. The structural basis regarding the gating regulation of CALHM1 remains mysterious. These are partially due to the unresolved N-terminal domain due to the flexibility of CALHM1, which is supposed to exist possibly in the pore. In this manuscript, Syrjänen et al. have reported the cryo-EM structures of human and chicken CALHM1 (hCALHM1, chCALHM1) for comparison. They solve the mutant hCALHM structures where RuR is bound or non-bound and present that the density in the middle of the pore surrounded by the pore lining N-terminal helices (NTH) is assigned to be RuR. Because of a physical block on the pore pathway, this structure is supposed to be closed or non-permeable. MD simulation is accompanied to show that the hydrophobic pocket prefers a phospholipid rather than cholesterol, and this lipid binding facilitates the stability of the CALHM1 channels. In general, the structural determination is reliable as there are few unreasonable parts in image analysis and modeling. The combination of patch-clamp recordings and MD simulation along with cryo-EM is a preferable strategy for understanding the structural basis of membrane channel proteins. While the visualized NTH and RuR in CALHM1 are interesting, I should admit that there are concerns regarding the authors' research methods, interpretation, and novelty of the structures. The authors should address the following issues I raised to make this manuscript more persuasive and comprehensible.

Major concerns:

1. It is described that the ordered NTHs in an upright manner would maximize the pore size and ion permeability. However, NTH is visible only when RuR is added to the I109W mutant. Isn't this a blocked structure without any activity? This structure does not guarantee that the upright NTHs make an open state unless the structure with NTHs surrounding the pore without RuR is determined. Because the RuR-free structure of hCALHM1I109W Δ ct does not show NTH, this assertion is not reasonable. In addition, the RuR bound hCALHM1 has the mutation of I109W, not wild type (WT). The authors show that I109W upregulates the channel activity (Fig. 3b), but it could be an artificial effect by the mutation, and the same is true for the RuR blockade because the structure of hCALHM1 Δ ct shows no density for NTH nor RuR. Given that, it is unclear what foundation Fig. 6 is drawn. The Fig.6 right panel stating "RuR plugging open channel" is wrong as a claim, and this is for Δ ct and I109W, not WT. The left of Fig. 6 is even more unsubstantiated.

- 1) We agree with the reviewer that calling the upright NTH bound to RuR the open channel is an overstatement. We revised the text accordingly and Figure 6 by removing the left hand side panel. We changed the right hand side panel to say "RuR plugging CALHM1 channel", removing the reference to what state (open or closed) the channel may be in.
- 2) We agree with the reviewer that it is ideal to capture RuR in the context of the WT hCALHM1. However, we faced technical challenges and had to compromise with the I109W mutant. Having said that, we showed in Fig. 5 (panel B) that RuR can inhibit the hCALHM1I109W Δ ct mutant therefore, the mutant serves as a tool to understand RuR binding.
- 3) The cryo-EM density for RuR is observed in the presence of RuR in the protein sample. Our hCALHM1 Δ ct structure (without the Ile109Trp point mutation) was solved in the absence of RuR, so we do not expect to see RuR density in this structure.

We revised the manuscript to more clearly deliver the above points.

2. The authors assigned RuR on the middle pore density. However, there is an ambiguity regarding the densities for RuR and NTHs in the current cryo-EM map, which does not reveal the feature of side chains of NTH nor make sure of the RuR orientation. The foundation of this assignment is that the oblong pore density corresponds to the size of RuR, and it disappeared in the RuR-free structure of hCALHM1I109W Δ ct (Line 219). In the RuR-free structure of hCALHM1I109W Δ ct, the densities assumed to be NTH along with RuR are lost. This is not enough to assign the orientation of RuR or the interacting side chains of hCALHM1I109W Δ ct with RuR. What if an unresolved peptide portion of hCALHM1I109W Δ ct perhaps contributes to these densities? The decreased inhibition by Q10R109W and Q13R109W (Fig. 5c, d) does not necessarily prove the repulsive effect against RuR nor that those two residues have direct interactions with RuR to fix RuR in the middle of the pore. Indeed, the partial inhibition of activity is still observed in those mutants (Fig. 5d). It is also not convincing that the concentration of RuR differs between 20 μ M in Fig. 5d and 50 μ M for structure determination. The assignment of RuR based on indirect evidence may cause more confusion.

1) We agree with the reviewer that the cryo-EM density is not sufficient to pinpoint the exact placement of RuR as we also indicated in our original manuscript even after extensive focused 3D classification around the pore region. We think that there are a number of subtly different binding modes within this binding pocket since the pore diameter is large. Furthermore, as RuR is a chemically symmetric molecule that is longer in one dimension than the other, the orientation of RuR is evident.

2) We see the oblong cryo-EM density consistent with the size of the RuR molecule observed in the middle of the pore only when RuR was added to the protein sample. The cryo-EM density does not accommodate a peptide. Furthermore, the pore is surrounded by hydrophilic residues from the eight subunits; thus, the environment would be challenging to place a peptide, hydrophobic compounds, or lipids.

3) The partial effect of the pore residue mutants (e.g. Q13R) is consistent with the binding mode that involves a number of residues and potentially multiple modes of bindings. We could not test double or triple mutants because mutating many of these pore residues changes the channel properties (loss of function) as for many other studies on pore residues and channel blockers.

4) Like many other structural studies aiming to capture compound binding, we tried to saturate the binding site of ruthenium red (RuR) by adding a high concentration of RuR to effectively visualize the compound. 50 μM was the highest concentration we could use without affecting image quality in cryo-EM. It is highly unlikely that the structure differs between 20 μM (Figure 5, Panel D) and 50 μM .

3. Related to the above, the authors use the C-terminal deletion mutant (Δct). In Ext. Data Fig. 5, no functional data for the full-length WT is shown, and hCALHM1 Δct is deemed WT, which is confusing. In this manuscript, no data for the full-length WT is shown, and all interpretations are based on hCALHM1 Δct . How can we be sure that this is physiologically significant, not an artificial effect? The authors presented that the C-terminal portion may facilitate an octameric assembly, but 9-mer itself is the consequence of Δct , which does not happen to the native CALHM1. Has the importance of the deleted C-terminus in determining oligomeric number been generally shown for other CALHMs subtypes, and how biologically significant is it?

The voltage-dependent ion channel activity of full-length CALHM1 has been shown in previous papers mainly by the Foskett's group and also our group (Syrjanen et al., 2020). This is the first manuscript to show that human CALHM1 Δct retains voltage-gated ion channel activity. The previous work on killifish CALHM1 (Demura et al., 2020) published a structure without the CTD and demonstrated that the ATP-permeation activity remained. Although they exclusively suggested 8-meric assembly in their paper reanalysis of their data (deposited to EMPIAR; #10444) revealed the presence of 9-meric species as well. We are therefore confident that CALHM1 Δct retains the functional profile of WT protein. The effect of truncating the CTD has not been shown for other CALHMs (2,4-6). Indeed the entire CTD has yet to be resolved in any of the published CALHM structures of any subtype. As the reviewer suggested, we avoided calling CALHM1 Δct the WT.

4. The authors demonstrated in MD simulation that the phospholipid binding in this hydrophobic pocket stabilizes the CALHM1 channel structure. Have the authors examined the calculation starting from the model without lipids in the hydrophobic pocket? How unstable the CALHM1 channel is when nothing occupies the hydrophobic pocket?

We have now performed additional all atom MD simulations with a model containing the NTHs. Histograms of $\text{C}\alpha$ atom RMSD values for the hCALHM1 ΔCT +NTH model show a rank order of stability of POPC > cholesterol > apo systems. In our original simulations we also observed that a channel starting from an 'apo' state exhibits instabilities, with larger drifts in RMSD compared to the POPC

bound structure that fail to converge over the course of the simulation. The new MD simulations results are presented in the revised Figure 5.

5. Lipid binding hydrophobic pocket.

a) Line 189, "On the other hand,.... the Val192Trp mutant likely stems from protein instability and a lack of trafficking to the plasma membrane."

Why do the authors say that only V192W lacks stability and membrane trafficking? In Fig. 3d, the signals for I109W and A199W are also weaker than for WT. It is likely that the mutants of I109W and A199W somehow lose the protein stability or normal trafficking if this analysis ensures quantitatively. Furthermore, only three mutants, not all studied, are shown in Fig. 3b. This representation is biased. Another confusing point in Fig. 3 is that the method to investigate surface expression is indirect. If possible, fluorescence microscopy is a more straightforward way to understand membrane transport. Is there any reason not to do so?

The suggested microscopic assessment for cell surface expression requires a well-behaving antibody that recognizes the extracellular region of CALHM1. No such antibody is commercially available.

Therefore, we implemented the described biochemical approach in this work.

Our original goal in the surface expression assay was to focus on the two mutations that had statistically significant upregulation or downregulation of activity compared to hCALHM1 Δ ct as measured by current density at 100 mV. These mutants were hCALHM1 Δ ct Ile109Trp (upregulation) and hCALHM1 Δ ct Val192Trp (downregulation). However, we appreciate the reviewer's suggestion to compare the surface level expression of all mutants. We therefore performed this assay on all mutants, analyzed the levels of surface expression by western blot and have included these data Figure 3d.. To summarize our findings, consistent with our FSEC analysis, hCALHM1 Δ ct Ala199Trp, hCALHM1 Δ ct Ile109Trp and hCALHM1 Δ ct express at higher levels compared to the other mutants. Indeed, consistent with our FSEC analysis, hCALHM1 Δ ct Ala199Trp has the highest expression. The other mutants Leu67Trp, Val112Trp, Ala116Trp, Thr196Trp, and Val192Trp express at multiple-fold lower levels on the cell surface.

-b) Fig. 3 The authors focused on 7 mutants around the hydrophobic pocket and generated Trp substitution mutants. The interpretation of Fig. 3b (Line 167, "First, incorporating the Val192Trp mutation.....") is incomprehensible. Looking at the structure model, the mutants of V112W, A116W, and A199W can cause the steric hindrance against lipids as interpreted for V192W, but it is noted that these tryptophan mutants are not positioned to strengthen or weaken hydrophobic contacts in the pocket. Why is it considered that only V192W causes pocket destabilization and the other three mutants do not?

We thank the reviewer for the comment and the thorough assessment of our structural coordinate. The A199W, V112W, and A199W can cause some steric hindrance but not as robust as V192W as in the mutant figure below.

-A199W (panel a) can be placed favorably without moving the main chain and this mutant has a higher surface expression level than the WT.

-V112W (panel b) can be placed without moving the main chain but it is in more crowded area compared to A199W. The surface expression was reduced, yet, the ion channel activity was still present.

-A116W (panel c) can be placed without moving the main chain but it is in more crowded area compared to A199W. The surface expression was reduced, yet, the ion channel activity was still present.

-V192W (panel d) cannot be placed without moving the main chain. V192W is in the deepest position of the pocket surrounded by residues such as L120, W189, L193, and V63. Thus, more robust effects compared to other mutants (e.g., reduced expression level and current density) is consistent.

We agree with the reviewer that the above points were not intuitive from our figure and not explained in the text. We addressed the above points in the revised manuscript.

a

b

c

d

-Finally, are these residues related to Alzheimer's disease?

No, these mutations are not related to the known Alzheimer's disease mutation.

-c) Line 205,

"Our structural analysis revealed....Ile109Trp 'cap' the hydrophobic pocket..."

The lipid density is also found in the structure of hCALHM1 Δ ct, suggesting that the I109 cap is not necessary for lipid binding. This should be addressed.

We agree.

- 1) We have revised the phrasing in the text. The reviewer is correct in stating that the Ile109Trp cap is not inevitable for the lipid binding but rather strengthen the lipid placement.
- 2) Importantly, we performed additional MD simulations on human Ile109Trp CALHM1 Δ ct for this revision. These data strongly support our hypothesis of increased lipid stability. The PMF calculations from these data indicate that POPC binding to human Ile109Trp CALHM1 Δ ct is even more energetically favorable, compared to human CALHM1 Δ ct. This new experimental result is incorporated into Figure 4d and the text in the revised manuscript.

Minor:

-The authors have reported the structure of chCALHM1 in nanodiscs in 2020 (Syrjänen et al. NSMB (2020)). What is the novelty in the RuR-free CALHM1 structures in this manuscript? Is there a significant difference? This should be addressed.

We agree that the points raised by the reviewer were not clearly described in our original manuscript.

- 1) The novelty of RuR-free CALHM1 structure is that it represents the first structure of a mammalian CALHM1 (human). There were no significant differences in major features such as the pattern of oligomeric assembly; however, it is the first study to show such structural conservation.
- 2) In this study, we were able to resolve regions of the protein not seen in our earlier cryo-EM model/map PDB-6VAM/EMD-21143. These include: the loop between TMD2 and TMD3 (residues 85-90, which reveals the position of the mutation involved in Alzheimer's Disease), part of the extracellular domain (residues 139-145) and an extended C-terminus (residues 249-260), which were not resolved in PDB-6VAM/EMD-21143.

We revised the manuscript to more clearly deliver the above points in Figure 1c, the architecture of the protomer is conserved. In addition, using the method of preparing chCALHM1 shown in this paper.

-Fig. 3b The error bars for I109W and T196W are pretty long. Is there any reason for the large variability?

This variability can be attributed to sources such as transfection efficiency or cell viability, which can be different between different cells. The datasets were collected over the course of several months on

different batches of HEK293 cells. We would also like to clarify that the graph referred to by the reviewer (Figure 3, Panel B) has the whiskers representing the minimum and maximum values for each mutant. As such, the top whisker for Thr196Trp reflects a single outlier data point. The boxes represent the median, 25th, and 75th percentile values.

To verify that the data is a single outlier, we have conducted additional patch-clamp experiments to increase sample sizes. The updated sample size (n) of each mutant is shown below and has been updated in Figure 3b and Supplementary Table 2.

WT CALHM1 Δ ct (n=10)

Ile109Trp (n=11)

Leu67Trp (n=7)

Val112Trp (n=5)

Ala116Trp (n=9)

Val192Trp (n=6)

Thr196Trp (n=10)

Ala199Trp (n=9)

For CALHM2, a different binding site of RuR has been shown. Is there any relevance to this study?

In short, the location of the published CALHM2 RuR binding site is not relevant to this study for the following reasons:

- 1) CALHM1 forms an octamer and CALHM2 forms an undecamer. As such, the pore sizes of CALHM1 and CALHM2 are drastically different, with CALHM2 having a much larger pore diameter. Thus, we do not expect to see the same site of association with ruthenium red (RuR).
- 2) As we discussed in our recent review of the large-pore channel field (Syrjanen and Michalski et al., 2021, JBC), cryo-EM density for the assigned binding site of RuR in the structure of CALHM2 is also present in the RuR-free structure. This density is consistent with CALHM2-Phe39, suggesting that it may have been misinterpreted and therefore, mis-assigned.

Reviewer #3 (Remarks to the Author):

This manuscript describes the structure of human CALHM1. This group reported the first cryo-EM structures of CALHM channels a couple of years ago, including chicken CALHM1. Novel observations here: identification of phospholipid binding in the structure (albeit in the same site previously identified by others as occupied by hemi-cholesterol; and observed also previously in other CALHM structures) with MD simulations suggesting that it helps to stabilize the channel structure. Second, observation of the channel inhibitor ruthenium red (RuR) in the channel pore, although another group previously demonstrated such a structure, albeit in a different CALHM channel paralog and with binding in a different site.

The work is well done, but fundamental new insights into the structural bases for CALHM channel

permeation and gating CALHM channel gating have not been provided that go beyond what these and other groups have previously reported. In particular, the orientation of the Nths with relationship to gating and permeation is still speculated upon, as it has been by other groups, without new hard data to support the model in Fig 6. My comments below touch on some of these aspects.

1. The authors suggest that the I109W mutation stabilized the Nth in the upright position, but then suggest that RuR is responsible for this structural placement. Furthermore, Nth seems not to be resolved in such a position in the RuR-free structure (Ext Fig 10). The authors need to clarify that they are speculating in either case. It could be that it is the RuR that stabilizes the structure and/or the mutation that does that. The former seems more likely.

We agree that we did not clarify this well in our original manuscript. In short, the upright position of NTH is stabilized by both the Ile109Trp mutation and RuR. 1) The structure of CALHM1 Δ ct Ile109Trp without RuR does not clearly resolve NTH indicating that there remains conformational flexibility of NTHs without RuR, 2) The structure of CALHM1 Δ ct Ile109Trp with RuR has more resolved NTHs. This is mediated by the lipid-Ile109-Trp-TMD1-NTH interaction and NTH-RuR interaction. These points are addressed more clearly in the revised manuscript.

2. The authors demonstrate that RuR inhibits channel currents, and they provide a structure of a mutant channel in the presence of RuR. A couple of things should be addressed by the authors.

First, why did the authors not solve the structure of WT CALHM in the presence of RuR...this might also help to address the previous comment.

Second, the authors need to comment on the environment surrounding the RuR in the pore. Does water surround the RuR? Does RuR manage in this structure to sterically block a 16Å diameter pore to small ions? Intuitively, this seems unlikely, but what does the structure say? How do the authors know that THIS structure is one of a “blocked” channel?

Another study demonstrated RuR binding to a different site in the channel (albeit a different CALHM). Notably, the Q10 and Q13 R mutations did not abolish RuR inhibition. Of note, the inhibition in the example trace for Q10R is inconsistent with the dot plot summary; and the currents for the Q16R channel in the absence of RuR seem different from WT channel behavior. Are the current densities for the Q-to-R mutants different from WT in the absence of RuR?

- 1) Our answer to the reviewer's first question: we agree that it would be more ideal to obtain CALHM1-RuR in the wildtype context. We made attempts to do so; however, we were unable to resolve RuR when the WT CALHM1-RuR was subjected to the cryo-EM study. As we mentioned above, the NTHs are conformationally flexible. Therefore, capturing RuR by cryo-EM may require an assistance of the Ile1109Trp mutation to stabilize the NTH.
- 2) Our answer to the reviewer's second question: we cannot see highly ordered water molecules. The local resolution is not sufficiently high to visualize ordered water or ions if present. Given the conformational flexibility of NTHs, it would be a technically challenging task. If we arbitrarily place side chains of the pore lining glutamines (please see the figure below), we measure the distances between RuR and the tip of glutamine residues to be 5.1 and 6.7 Å. Therefore, we could only say that there is a low chance that water can sit between RuR and the pore-lining

glutamines. Nevertheless, based on our structure-based site directed mutagenesis results in Figure 5, we think that this RuR-bound CALHM1 structure represents the blocked state.

- 3) Thanks for pointing it out. A better representative trace for the Q10R_{1109W} recording is now shown in Figure 5. Current traces obtained from the Q16R_{1109W} mutant are indeed different from WT_{1109W}. This mutant channel exhibited slower activation kinetics, which could be related to the location of this residue at the bottom of the pore.

In the below figure, current density-voltage relationships for all the Gln-to-Arg point mutations studied are shown.

As shown, the Q13R_{1109W} channel is the only one that exhibits a significant reduction in current density compared to WT_{1109W} (two-tailed unpaired t-test, Welch's correction), reaching statistical significance for current densities at +20 ($p = 0.009$), +40 ($p = 0.004$), +60 ($p = 0.006$), +80 ($p = 0.016$), and +100 mV ($p = 0.017$).

The article that suggested a different binding site of RuR for CALHM2 is not pertinent to this study. We do not expect to see the same site of association with RuR as CALHM1 and CALHM2 have drastically

different pore diameters. As we discussed in a recent review (Syrjanen and Michalski et al., 2021, JBC), cryo-EM density for the assigned binding site of RuR in CALHM2 is also present in the RuR-free structure. This density is consistent with density for Phe39, suggesting that it may have been incorrectly modeled.

3. Please clarify how structures of I109W-CALHM1 were obtained in the absence of RuR. The text states that it was not possible because of toxicity, yet data are shown and referred to.

We have clarified this in the methods. In brief, human Ile109Trp calhm1 Δ ct was expressed in HEK GnTI- cells with 20 μ M RuR. After binding to Strep-Tactin resin, the protein was extensively washed with CaCl₂ (20 mM), EGTA (5 mM), and EDTA (40 mM) to remove RuR.

4. The currents in Ext Fig 5 should be presented as current densities.

As requested by the reviewer, we have changed the axis on the scale bar in Supplementary Figure 5 to reflect current densities.

5. Regarding the I109W effects on currents. It's not obvious from the description of the structure why only this mutation enhanced the currents. It is surprising that the authors did not extend their MD simulations to this mutant to confirm structural stabilization. In addition, how can the authors know that a "capping and stabilizing" effect accounts for the enhanced current density?

Our analysis of the cryo-EM map of human Ile109Trp calhm1 Δ ct with and without RuR reveals density for a phospholipid in the conserved hydrophobic pocket, as in the human calhm1 Δ ct structure. We also observe density for Ile109Trp, and suggest that Ile109Trp "caps" the hydrophobic pocket. From our structural analysis, we speculate this capping contributes to enhanced current density.

We have also performed additional MD simulations to examine the binding energy of POPC in the hydrophobic pocket formed by human Ile109Trp calhm1 Δ ct. (revised Figure 4d). The PMF calculations from these data indicate that POPC lipid binding to human Ile109Trp CALHM1 Δ ct is more energetically favorable than POPC binding to human CALHM1 Δ ct.

Reviewer #4 (Remarks to the Author):

The manuscript reports on a structural investigation of the voltage-dependent channel CALHM1. cryoEM, electrophysiology, and molecular dynamics simulations are used to clarify some aspects of channel activity and its modulation, the structure of the pore and the binding site of ruthenium red, known to inhibit the channel. The results are original and, given the mechanistic insight they provide, of interest to a wide community of scientists. The conclusions are in general supported by the data. In particular, the molecular simulations are competently performed and properly analyzed: the arguments presented in support of the hypothesis that the pocket prefers POPC over cholesterol are compelling. Overall, I am enthusiastic about the paper and I recommend publication. There are only few clarifications and comments that I recommend:

1) I am a bit unclear as to whether or not the NTH has been resolved in all the structures or only in the RuR-bound one (my understanding is that the latter is true). I suspect that the information is reported

somewhere, but I suggest the authors to specify more clearly which residues were modeled in each structure.

The understanding of the reviewer is correct: the NTHs are only resolved in the RuR-complexed structure. To clarify which N-terminal residues are modeled in each structure as requested by the reviewer, we have added this information to a new Supplementary Figure 11b. We also summarize the most N-terminal residues modeled in each structure here:

Chick calhm1 Δ ct: Ala27

human calhm1 Δ ct: Ala30

human calhm1 Δ ct Ile109Trp with RuR: Met1

human calhm1 Δ ct Ile109Trp without RuR: Gly25

2) Related to the first question: what model was used for the MD simulations? Was the NTH present? How about RuR, was it modeled?

The models used were: human CALHM1 Δ ct (using the hCALHM1 Δ ct model in which the NTHs were not resolved), human CALHM1 Δ ct Ile109Trp (using the model of the RuR-CALHM1 Δ ct Ile109Trp structure that included the NTHs. RuR was not modeled in the simulations), and human CALHM1 Δ ct with the NTHs included (this model was created by reverting Trp109 from the RuR-CALHM1 Δ ct Ile109Trp structure to isoleucine). Each model was examined in the context of either cholesterol or POPC for coarse grained systems, for a total of 6 PMF calculations. RuR was not modeled for any simulations.

3) If NTH was included in the simulation, was any insight gathered about the dynamics of this helix in presence and in absence of RuR? Do these observation support the proposed mechanism shown in Fig 6?

- 1) Our original submission did not contain NTH. We now ran a new set of MD simulations containing the NTH (without RuR) and found that the binding preference of POPC over cholesterol in the 'lipid binding pocket' is retained.
- 2) Additionally, we ran a new set of MD simulations on hCALHM1 Δ ct Ile109Trp. Our PMF calculation indicated that POPC binds more stably in hCALHM1 Δ ct Ile109Trp than hCALHM1 Δ ct.

The above results are shown in Figure 4d of the revised manuscript.

4) A crucial question regarding the lipid binding pocket is the specificity for particular lipid species. For instance, the channel might bind selectively unsaturated or saturated lipids. Which one of the two lipid tails (palmitic or oleic) is in contact with TMD1 and NTH? Did the authors simulated alternative binding poses with one or the other lipids tail in contact with NTH?

At the current cryo-EM map resolution of the lipids, most likely due to the highly mobile nature of the lipid tails, we cannot define whether the palmitic or oleic tail is in the proximity of TMD1/NTH. We agree with the reviewer that specificity of the hydrophobic pocket for particular lipid species is an intriguing

area of study for future work .The large number of torsional degrees of freedom for each lipid tail makes performing an analysis of this type on atomistic simulations outside the scope of this current paper.

REVIEWERS' COMMENTS

Reviewer #1 (Remarks to the Author):

The resubmitted version of the manuscript includes new molecular dynamics simulations supporting the role of Isoleucine 109 as a lipid-binding residue and lipid/protein stability. Additionally, the authors have checked for plasma membrane expression of all the mutants used in electrophysiological experiments, which validate further their interpretations. Although the authors could not provide a direct demonstration for the role of lipids in CALHM-1 channel activity, the data support a putative conserved lipid-binding domain on channel function. I understand, however, that a solid demonstration on how lipids regulate CALHM-1 channel function is quite challenging and alone could be a new manuscript. Overall, most of the major concerns have been addressed by the authors. I just have some minor concerns/comments detailed below:

Minor Concerns:

- 1) Line 26-28: "We demonstrated... that the phospholipid binding stabilizes the channel structure and regulates the channel activities". Please tone down this statement in the abstract section. The evidence only supports a putative role of phospholipids in channel stabilization and potentially in channel function (e.g., gating properties). In my view, there is not yet a solid/direct demonstration that lipids can regulate channel activity.
- 2) Line 115-116 "This pocket is formed mostly by the hydrophobic residues, which are conserved among the CALHM1 orthologs (Fig. 2a)". The conservation of the hydrophobic residues described for the lipid-binding pocket is not shown in the manuscript. The readers could benefit if the authors incorporate a sequence alignment to show the conservation of these residues with other orthologs.
- 3) Supplementary Fig. 4 Figure is still confusing. Grey beads are indicated to be POPC and cholesterol, which seems wrong; please check this. In addition, the magnification in panel B does not seem to match the selected area. It is also not clear that assigned lipid locations in the figure corresponds to the proposed lipid pocket.
- 4) Result section: Description of the western blot data is redundant: Two statements made in Lines 159-163 and 176 – 178 repeat the same idea and have inconsistency describing the effect of Ile109Trp mutation on protein expression. Please, check.

5) Lines 208-209: “The simulations also demonstrated that cholesterol binds more favorably to the hCALHM1I109W Δ ct than hCALHM1 Δ ct (-1.45 kcal/mol in hCALHM1I109W Δ ct vs +5.5 kcal/mol in hCALHM1 Δ ct)” This statement is not consistent with what is shown in the corresponding figure. Please, check and correct this statement.

6) Since the authors refer sometimes to hCALHM1 Δ ct as WT, I strongly recommend being consistent along the text and figures. If possible, use the same abbreviation, to avoid confusion.

7) There is a discrepancy in the Western blot showing a clear band for V112W in the biotinylated group but no band for the same mutant in the total expression; please check.

8) Line 363. Please add that *** denotes $p < 0.001$ versus WT. Same observation for Fig. 6D: Please add in the figure legend which group was used for comparisons.

9) Supplementary Table 4. For the paired-T test group, it is not clear what group was used for comparison. Is the amplitude values expressed as a percentage? Please clarify.

Reviewer #2 (Remarks to the Author):

The authors have greatly improved the manuscript.

Western blotting results have been incorporated, making the interpretations in terms of membrane localization and its efficacy more convincing.

The additional MD simulations support that I109W allows the binding of lipids more tightly.

Since the RuR assignment differs from another study on CALHM2 (Choi et al. Nature, 2019), it may confuse the field to claim an ambiguous alternative assignment definitively.

Unless the structure of hCALHM1 WT with RuR is clearly determined, it could be an I109W-specific observation. Whether RuR binds to and blocks the center of the pore in WT is not experimentally clarified.

Currently, I do not strongly disagree with the RuR assignment by the authors, but in the unlikely event that this assignment could be at risk depending on future studies.

It is recommended to mention in the Discussion section something about the possibility that this may not apply to WT hCALHM1 or that it may not be physiological.

I would also like to leave a couple of minor issues.

1) Is there any biological significance in stabilizing POPC binding by the I109W mutant?

2) Are Figure 4d right plot colors of WT (blue) and I109W (green) opposite? From the text, I109W should indicate a lower PMF (Line 208 “The simulations also demonstrated that cholesterol binds more favorably...”).

Reviewer #3 (Remarks to the Author):

The authors have addressed the concerns.

Reviewer #4 (Remarks to the Author):

All my concerns and requests for clarification have been appropriately addressed. I recommend publication of the manuscript in the present form.

We thank the reviewers again for the comments and suggestions on our revised paper. We considered all comments made by Reviewers 1 and 2 and further revised the paper accordingly. Responses to the reviewers' comments are in blue.

REVIEWERS' COMMENTS

Reviewer #1 (Remarks to the Author):

The resubmitted version of the manuscript includes new molecular dynamics simulations supporting the role of Isoleucine 109 as a lipid-binding residue and lipid/protein stability. Additionally, the authors have checked for plasma membrane expression of all the mutants used in electrophysiological experiments, which validate further their interpretations. Although the authors could not provide a direct demonstration for the role of lipids in CALHM-1 channel activity, the data support a putative conserved lipid-binding domain on channel function. I understand, however, that a solid demonstration on how lipids regulate CALHM-1 channel function is quite challenging and alone could be a new manuscript. Overall, most of the major concerns have been addressed by the authors. I just have some minor concerns/comments detailed below:

Minor Concerns:

1) Line 26-28: “We demonstrated... that the phospholipid binding stabilizes the channel structure and regulates the channel activities”. Please tone down this statement in the abstract section. The evidence only supports a putative role of phospholipids in channel stabilization and potentially in channel function (e.g., gating properties). In my view, there is not yet a solid/direct demonstration that lipids can regulate channel activity.

Our MD data suggests that the hydrophobic pocket has specificity towards which lipids are energetically favored to bind. Indeed, these data suggest that binding of a phospholipid, which we observe density for in the hCALHM1 cryo-EM structure, is more favorable than that of cholesterol. By perturbing the nature of the amino acids in the hydrophobic pocket through point mutations, we discovered that lipid binding stability, channel activity as well as channel stability were affected. This leads us to suggest that phospholipid binding does indeed stabilize the channel and has an effect on channel activities.

We toned down and re-wrote the sentence as follow: “We demonstrate by MD simulations that this pocket preferentially binds phospholipid over cholesterol to stabilize its structure and regulates the channel activities. Finally, we show that residues in the amino-terminal helix form the channel pore that ruthenium red binds and blocks.”

2) Line 115-116 “This pocket is formed mostly by the hydrophobic residues, which are conserved among the CALHM1 orthologs (Fig. 2a)”. The conservation of the hydrophobic residues described for the lipid-binding pocket is not shown in the manuscript. The readers could benefit if the authors incorporate a sequence alignment to show the conservation of these residues with other orthologs.

We have now incorporated a sequence alignment into Supplementary Figure 4, panel a. The regions of human, mouse, chicken, zebrafish and killifish CALHM1 that encompass the residues of the hydrophobic pocket are indicated in the alignment by asterisks.

3) Supplementary Fig. 4 Figure is still confusing. Grey beads are indicated to be POPC and cholesterol, which seems wrong; please check this. In addition, the magnification in panel B does not seem to match the selected area. It is also not clear that assigned lipid locations in the figure corresponds to the proposed lipid pocket.

We have modified Supplementary Fig. 4 to improve clarity. Additionally, we have included three Supplementary movies which also support the information in this figure.

4) Result section: Description of the western blot data is redundant: Two statements made in Lines 159-163 and 176 – 178 repeat the same idea and have inconsistency describing the effect of Ile109Trp mutation on protein expression. Please, check.

Lines 159-163 describe levels of protein expression of hCALHM1 Δ ct and the point mutations, as analyzed by western blotting. Lines 176-178 discuss surface expression in light of effects on channel activity. The sentence has been amended to correct the effect of the Ile109Trp mutation on surface level expression.

5) Lines 208-209: “The simulations also demonstrated that cholesterol binds more favorably to the hCALHM1I109W Δ ct than hCALHM1 Δ ct (-1.45 kcal/mol in hCALHM1I109W Δ ct vs +5.5 kcal/mol in hCALHM1 Δ ct)” This statement is not consistent with what is shown in the corresponding figure. Please, check and correct this statement.

We have amended the text so that it accurately reflects the content in the figures.

6) Since the authors refer sometimes to hCALHM1 Δ ct as WT, I strongly recommend being consistent along the text and figures. If possible, use the same abbreviation, to avoid confusion.

We agree. We have clarified this in the text and figures.

7) There is a discrepancy in the Western blot showing a clear band for V112W in the biotinylated group but no band for the same mutant in the total expression; please check.

The band for V112W in the western blot for total expression is indeed faint. It is difficult to use longer exposure times or amend the contrast as then the bands corresponding to proteins with higher expression obscure the bands representing proteins which are poorly expressed in comparison.

We do not expect to be able to draw quantitative comparisons between two different blots, especially as one blot represents total protein and the other blot represents protein that has been enriched for surface level expression.

8) Line 363. Please add that *** denotes $p < 0.001$ versus WT. Same observation for Fig. 6D: Please add in the figure legend which group was used for comparisons.

We have modified both sentences according to the reviewer's suggestions. Line 363 now states "*** denotes $p < 0.001$ versus wildtype hCALHM1 Δ ct" and lines 415-416 say now "****, ***, and ** denote $p < 0.0001$, $p < 0.001$, and $p < 0.01$, respectively, versus basal conditions (absence of RuR) for each construct studied".

9) Supplementary Table 4. For the paired-T test group, it is not clear what group was used for comparison. Is the amplitude values expressed as a percentage? Please clarify.

As stated in our previous rebuttal, for the paired t-tests, comparisons were made between the amplitude after (in the presence of RuR) and before the treatment (in the absence of RuR) for each construct studied. For instance (see values in Supp. Table 4), the amplitude of Ile109Trp hCALHM1 Δ ct after RuR application was compared to the amplitude of Ile109Trp hCALHM1 Δ ct after before RuR application: the mean remaining current amplitude after RuR application was a $7.1 \pm 1.1\%$ of the original current amplitude (basal conditions, no RuR) and the statistic for this (before versus after) was $p < 0.0001$. The same was done for all the constructs studied.

The reviewer is correct in that amplitude values are stated as percentages of current amplitude (as stated in the title of the Supplementary Table: "normalized current amplitude"). To clarify this, we have substituted "Amplitude" for "Amplitude (%)" in the headings of the table.

Reviewer #2 (Remarks to the Author):

The authors have greatly improved the manuscript.

Western blotting results have been incorporated, making the interpretations in terms of membrane localization and its efficacy more convincing.

The additional MD simulations support that I109W allows the binding of lipids more tightly.

Since the RuR assignment differs from another study on CALHM2 (Choi et al. Nature, 2019), it may confuse the field to claim an ambiguous alternative assignment definitively.

Unless the structure of hCALHM1 WT with RuR is clearly determined, it could be an I109W-specific observation. Whether RuR binds to and blocks the center of the pore in WT is not experimentally clarified.

Currently, I do not strongly disagree with the RuR assignment by the authors, but in the unlikely event that this assignment could be at risk depending on future studies.

It is recommended to mention in the Discussion section something about the possibility that this may not apply to WT hCALHM1 or that it may not be physiological.

We agree with the reviewer that the field has been confused as we introduced and discussed it. 1) CALHM2 does not show ion channel activities in the hands of many, including us. Thus, the functional output of RuR inhibition in CALHM2 has been questionable to many. 2) In Choi et al Nature 2019, the authors claim RuR density in CALHM2 is also observed in the CALHM2 structure without RuR in the same paper. This discrepancy was addressed in a review by Syrjanen et al (JMB 2021).

This gave us (and the field) the reason to explore the true RuR inhibition mode on CALHM1 that forms a bona fide channel. In this study, we showed that the full-length CALHM1, hCALHM1 Δ ct, and Ile109Trp hCALHM1 Δ ct all retain voltage-gated ion channel activity. Furthermore, we have shown that, like full-length CALHM1 (as demonstrated via various papers by the Foskett lab as well as our group; Syrjanen et al., 2020), Ile109Trp hCALHM1 Δ ct is also inhibited by ruthenium red.

Little or no change is observed between the core structural components of the CALHM1 assembly (TMD2-4 and the CTH) in hCALHM1 Δ ct and RuR- Ile109Trp hCALHM1 Δ ct (RMSD=0.787 Å over 178 C α positions), suggesting that pore block by RuR involves the movement of TMD1 and NTHs.

Of course, it would be more ideal to determine the structure of RuR-hCALHM1 Δ ct. We were unable to do so because of technical challenges. Our structural and functional observations remain consistent with the suggestion that RuR blocks the central pore in hCALHM1 Δ ct. Indeed, it is difficult to develop an argument for inhibition by RuR of wild-type human CALHM1 to no longer occur via block of the central pore but through an alternative mechanism.

In the Discussion, we noted that RuR had been reported to bind to a different site in CALHM2. We have also noted in the Introduction that there is controversy about whether CALHM2 forms a voltage-gated ion channel.

I would also like to leave a couple of minor issues.

1) Is there any biological significance in stabilizing POPC binding by the I109W mutant?

The biological significance here is the mechanistic insights gained in using the Ile109Trp mutation as a tool to investigate the lipid binding properties of human CALHM1. As stated by the reviewer, Ile109Trp strengthens lipid placement in the hydrophobic pocket. By comparing PMF calculations of cholesterol and POPC in hCALHM1 Δ ct, we discovered that the pocket has specificity towards which lipids are energetically favored to bind. By perturbing the nature of the amino acids in the hydrophobic pocket through point mutations, including Ile109Trp, we discovered that lipid binding stability, channel activity as well as channel stability were affected. This leads us to the suggestion that in a biological context, phospholipid binding stabilizes the channel and has an effect on channel activities.

2) Are Figure 4d right plot colors of WT (blue) and I109W (green) opposite? From the text, I109W should indicate a lower PMF (Line 208 “The simulations also demonstrated that cholesterol binds more favorably....”).

Thanks for the comment. We have amended the text accordingly so that it reflects what is observed in Figure 4d.

Reviewer #3 (Remarks to the Author):

The authors have addressed the concerns.

Reviewer #4 (Remarks to the Author):

All my concerns and requests for clarification have been appropriately addressed. I recommend publication of the manuscript in the present form.